# Germinal center output is sustained by HELLS-dependent DNA-methylation-maintenance in B cells

Clara Cousu[1,6], Eléonore Mulot[1,6], Annie De Smet[1], Sara Formichetti[2,3], Damiana Lecoeuche[1], Jianke Ren[4,5], Kathrin Muegge [4], Matthieu Boulard [2], Jean-Claude Weill[1], Claude-Agnès Reynaud [1] & Sébastien Storck [1] ✉

HELLS/LSH (Helicase, Lymphoid Specific) is a SNF2-like chromatin remodelling protein involved in DNA methylation. Its loss-of-function in humans causes humoral immunodeficiency, called ICF4 syndrome (Immunodeficiency, Centromeric Instability, Facial anomalies). Here we show by our newly generated B-cell-specific *Hells* conditional knockout mouse model that HELLS plays a pivotal role in T-dependent B-cell responses. HELLS deficiency induces accelerated decay of germinal center (GC) B cells and impairs the generation of high affinity memory B cells and circulating antibodies. Mutant GC B cells undergo dramatic DNA hypomethylation and massive de-repression of evolutionary recent retrotransposons, which surprisingly does not directly affect their survival. Instead, they prematurely upregulate either memory B cell markers or the transcription factor ATF4, which is driving an mTORC1-dependent metabolic program typical of plasma cells. Treatment of wild type mice with a DNMT1-specific inhibitor phenocopies the accelerated kinetics, thus pointing towards DNA-methylation maintenance by HELLS being a crucial mechanism to fine-tune the GC transcriptional program and enable long-lasting humoral immunity.

Humoral immunity confers a non-redundant protection against extracellular bacteria and some viruses, as evidenced by the high susceptibility of patients with antibody deficiency to these microbes[1,2]. Antibody responses elicited by T-dependent (TD) protein antigens usually convey superior protection as T-cell help is crucial for efficient class-switch recombination and formation of germinal centers (GC). Indeed, B cells undergo massive proliferation and somatic hypermutation of their rearranged immunoglobulin genes in the GC dark zone (DZ), followed by selection for their antigen affinity in the GC light zone (LZ), which enables a gradual affinity maturation of

immunoglobulins (Ig)[3,4]. Moreover, GC is considered as the major source of high-affinity memory B cells (MBC) and long-lived plasma cells (LLPC) that support life-long protection[5,6], even though some low-affinity long-lived cells can arise in a GC-independent way[7]. Accordingly, the ability to form and maintain GCs is key to successful immunization[8]. Both the selection for antigen affinity, and the selection for MBC and LLPC fates rely on signals sent by follicular helper T cells and by the BCR itself, which remodel the B cell transcriptome and its metabolism through differential activation of the mTORC1 pathway and many transcription factors, notably c-MYC or IRF4[9,10].

[1]Université Paris Cité, CNRS UMR 8253, INSERM U1151, Institut Necker Enfants Malades, F-75015 Paris, France. [2]Epigenetics and Neurobiology Unit, European Molecular Biology Laboratory (EMBL), 00015 Monterotondo, Italy. [3]Joint PhD degree program, European Molecular Biology Laboratory and Faculty of Biosciences, Heidelberg University, Heidelberg, Germany. [4]Epigenetics Section, Frederick National Laboratory for Cancer Research in the Mouse Cancer Genetics Program, National Cancer Institute, Frederick, MD, USA. [5]Present address: NHC Key Lab of Reproduction Regulation,Shanghai Engineering Research Center of Reproductive Health Drug and Devices, Shanghai Institute for Biomedical and Pharmaceutical Technologies, Shanghai 200237, China. [6]These authors contributed equally: Clara Cousu, Eléonore Mulot. ✉e-mail: sebastien.storck@inserm.fr

However, despite recent advances, the factors that dictate GC duration and differentiation into MBCs and LLPCs are not fully elucidated.

Understanding the pathophysiology of natural antibody deficiencies may help uncover new factors that regulate B-cell activation and differentiation within GCs. The human ICF (Immunodeficiency, Centromeric Instability and Facial anomalies) syndrome consists in a rare inherited immunodeficiency often accompanied by mental retardation and facial dismorphy (e.g., hypertelorism, flat nasal bridge, epicanthus)[11–13]. About 80 ICF patients have been described so far, who suffer from recurrent, sometimes life-threatening infections of the respiratory and digestive tracts[13–15]. Almost all patients display hypo- or agammaglobulinemia, often with a concomitant decrease in IgG, IgA and IgM, and an absence of vaccine antigen-specific antibodies[16,17]. In addition, their CD27+ MBC compartment is most often drastically reduced even when they have normal peripheral B cell counts[16–19]. As a whole, this phenotype is consistent with an antibody deficiency consecutive to an impaired B-cell differentiation in secondary lymphoid organs.

On a molecular level, patients' phytohemagglutinin-stimulated peripheral blood lymphocytes present pathognomonic decondensation of heterochromatic pericentromeric regions of chromosomes 1, 9 and 16[11]. This decondensation is consecutive to hypomethylation of the repeated satellite sequences that compose these regions[20–22]. Hypomorphic mutations in *DNMT3B* (DNA methyltransferase 3B), that codes for one of the two de novo DNA methyltransferases, are found in 50% of the patients (ICF1)[23,24]. Mutations in three additional genes have been described[25,26]: the transcription factor ZBTB24 (Zinc finger, and broad complex, tramtrack, and Bric-à-brac 24, ICF2), the transcription regulator CDCA7 (Cell Division Cycle-Associated protein 7; ICF3), and the SNF2-family chromatin remodeler HELLS/LSH belonging to the superfamily of helicases (Helicase, Lymphoid Specific; ICF4)[27,28], a protein already known to be required for the establishment and maintenance of DNA methylation[29–33]. Initially, a model was proposed to integrate all four proteins in a common DNA methylation pathway: ZBTB24 would direct the transcription of CDCA7[34], which recruits HELLS to form a bipartite chromatin remodeler that would facilitate DNMT3B access to DNA for de novo methylation[35]. However, more recently, the CDCA7/HELLS complex was shown to promote DNA methylation maintenance of late-replicating regions in a replication-uncoupled manner, by allowing DNMT1 and its co-factor UHRF1 to access heterochromatic regions through its chromatin remodeling activity[36–40]. These regions are most notably found hypomethylated in ICF2, 3 and 4, compared to ICF1 patients[41], which underlines that DNMT3B on one side, and ZBTB24, CDCA7, and HELLS on the other, can play separate roles in de novo and maintenance DNA methylation respectively. In addition, recent data indicate that the CDCA7/HELLS complex is involved in double-strand break repair through non-homologous end-joining, which may affect class-switch recombination and account for some features of ICF cells[42,43].

To decipher the mechanisms at the origin of the antibody deficiency in ICF4 syndrome, and more specifically the role of HELLS and DNA methylation during antibody responses, we describe here a mouse model in which critical exons of *Hells* are conditionally ablated during early B-cell development (*Mb1*-Cre *Hells^{KO}*) or in mature naive B cells (*CD21*-Cre *Hells^{KO}*). We report that HELLS expression in B cells is mandatory for DNA methylation maintenance, which prevents the premature termination of the GC reaction and enables the establishment of a long-term humoral immunity.

## Results

### Conserved expression of *Hells* and *Cdca7* in human and mouse germinal center B cells

As ICF phenotype is suggestive of a B-cell intrinsic dysfunction, we first determined the pattern of expression of the four ICF genes in human and murine B-cell subpopulations. Publicly available RNA-seq

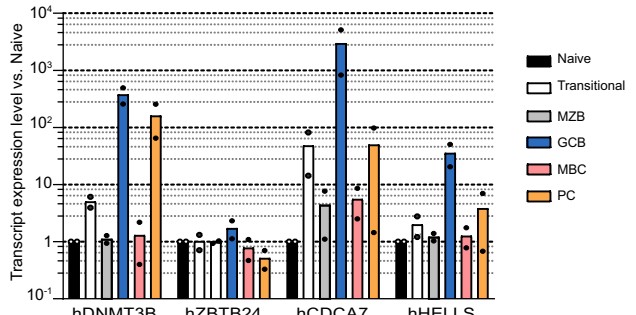

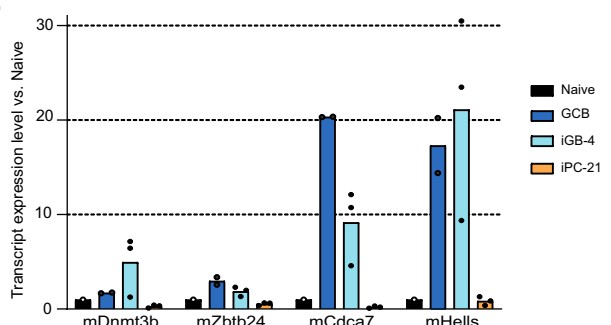

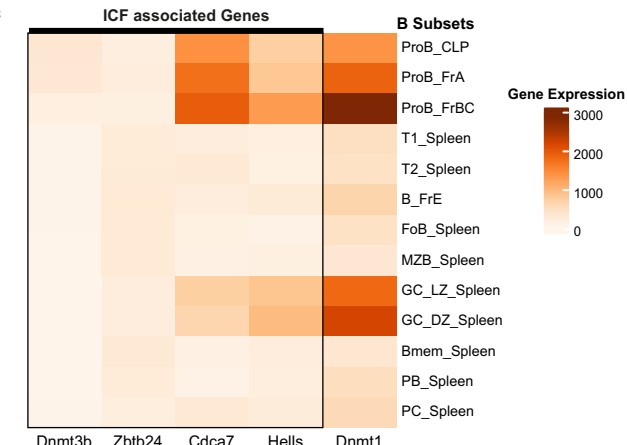

**Fig. 1 | *Hells* is highly upregulated in human and mouse during the germinal center reaction. a** Quantification by RT-qPCR of *DNMT3B*, *ZBTB24*, *CDCA7*, and *HELLS* transcripts of cells sorted from pediatric spleens (n = 2). Naive: CD19+CD38+CD24+IgD+IgM+CD27neg; transitional: CD19+CD38hiCD24hi; Marginal Zone B cells (MZB): CD19+CD38+CD24+IgD+IgM+CD27+; Germinal Center B cells (GCB): CD19+CD38+CD24neg; Memory B cells (MBC): CD19+CD38+CD24+IgDnegCD27+IgMneg; Plasma cells (PC): CD19+CD38hiCD24neg. **b** Quantification by RT-qPCR of transcripts of splenic naive and GCB cells from WT mice (n = 2), and in vitro induced GCB-like (iGB-4) and ASC (iPC-21) (n = 3), according to Nojima et al.[47]. **c** Heatmap of the expression of the four ICF genes and of *Dnmt1* at all stages of B-cell development and activation in the mouse. RNAseq data were collected from Immgen database GSE109125. Bar charts show relative fold-change of each transcript compared to its level in naïve B cells, after normalization to β2-microglobulin. Each dot represents one sample.

transcriptome datasets[44,45] combined with our real-time quantitative RT-PCR (RT-qPCR) analysis on pediatric spleens show that human *DNMT3B*, *CDCA7* and *HELLS* display a biphasic expression pattern: their expression is much higher in bone marrow B-cell progenitors than in immature and naive B cells, markedly re-induced at the GC stage and subsequently repressed upon terminal differentiation into MBC or PC (Fig. 1a and Supplementary Fig. 1a, b). In contrast, *ZBTB24* expression remains relatively constant throughout B-cell development and

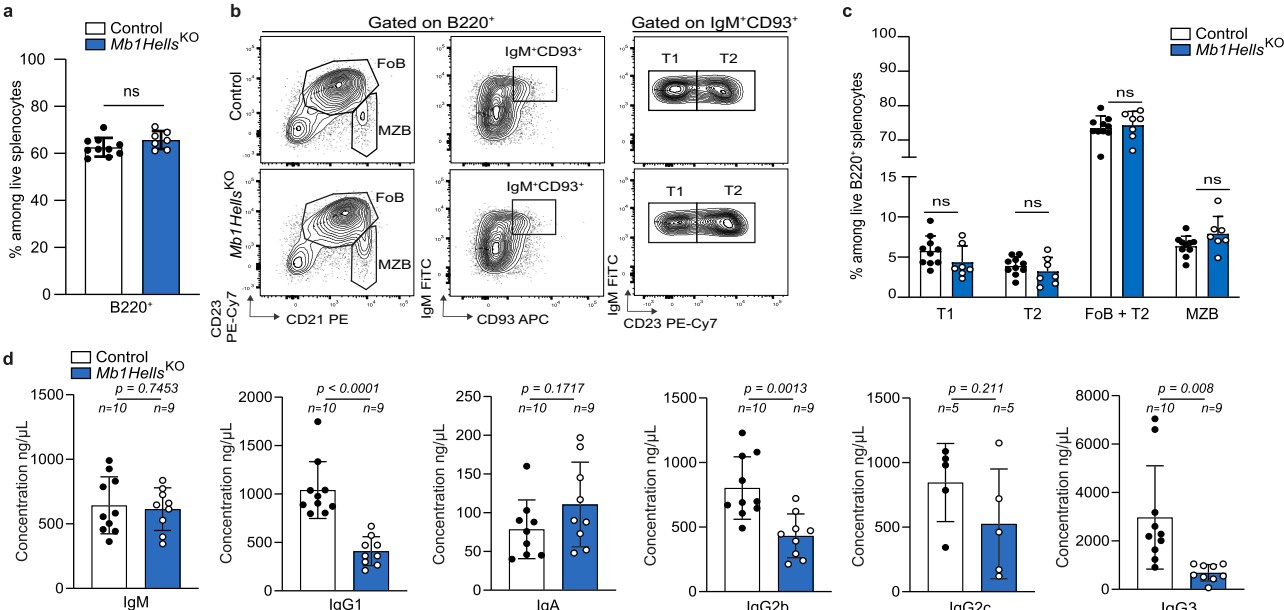

**Fig. 2 | Hells B-cell conditional knockout mice display normal B-cell development but reduced basal antibody titers. a** Percentage of B lymphocytes (B220⁺) among living splenocytes, measured by flow cytometry in control ($n = 10$) and $Mb1Hells^{KO}$ mice ($n = 7$). **b** Representative flow cytometry plots showing type 1 and 2 transitional B cells (T1: B220⁺IgM⁺CD93⁺CD23ⁿᵉᵍ; T2: B220⁺IgM⁺CD93⁺CD23⁺), follicular B cells (FoB): B220⁺CD23⁺CD21⁺; and marginal zone B cells (MZB): B220⁺CD21ʰⁱCD23ˡᵒ/ⁿᵉᵍ in the spleen of non-immunized control and $Mb1Hells^{KO}$ mice. **c** Quantification of the splenic subsets depicted in (**b**) in control ($n = 10$, of which 7

were $Mb1^{Cre/WT}Hells^{F/WT}$) and $Mb1Hells^{KO}$ mice ($n = 7$). **d** Quantification by ELISA of immunoglobulin titers of non-immunized 12-weeks old mice. Number ($n$) of control (of which 4 were $Mb1^{Cre/WT}Hells^{F/WT}$) and $Mb1Hells^{KO}$ mice used is indicated in the figure. Data were pooled from two independent experiments. Bar charts and error bars represent the mean ± SD. Unpaired two-tailed t-test (**a**) and (**c**) and two-tailed Welch's t-test (**d**) were performed, ns: non-significant, exact $p$ values indicated in figures. Source data are provided in Source Data File.

differentiation. Data from Immgen[46] and our own RT-qPCR experiments indicate that this dynamic expression pattern is conserved in the mouse, with a ≈8–20-fold and ≈10–17-fold upregulation of *Cdca7* and *Hells* in pro-B, GC B and in vitro-derived murine GC-like B cells (iGB4)[47] compared to naive B cells (Fig. 1b, c). A notable exception concerns *Dnmt3b*, whose transcription remains rather low in murine centroblasts and centrocytes (Fig. 1b, c). Collectively, these data point to a potential role of the ICF pathway during the proliferation steps of B-cell differentiation and activation, which is further supported by the predominant expression of *HELLS* and *CDCA7* in the cycling subpopulation of human tonsil GC B cells[48].

### *Hells* B-cell conditional knockout mice display normal B cell development

The conserved expression profile of *Hells* in B-cell progenitors and in GC B cells, and its well-established role in DNA methylation prompted us to delete this gene in the B-lineage to model ICF4 syndrome in the mouse. Floxed mice bearing an allele with *Hells* exons 9 and 10 flanked by LoxP sites (*Hells^F* allele, Supplementary Fig. 2a)[43] were crossed with the *Mb1-Cre* strain, which expresses the Cre recombinase specifically in the B-cell lineage from the pro-B cell stage[49]. The resulting offspring were inter-crossed and maintained on a mixed C57BL6:129 genetic background. The mutant animals described in this study were either homozygous for the floxed allele (*Mb1-Cre Hells^{F/F}*) or had a knockout allele generated by germline deletion (*Mb1-Cre Hells^{F/Δ}*); they were compared with control littermates devoid of Cre transgene (e.g. *Hells^{F/F}*, *Hells^{F/Δ}*), unless specified in the figure legends. For a subset of experiments, *Hells^{F/F}*-mice were crossed with *CD21-Cre*, in which the Cre recombinase is expressed from the T2 transitional stage[50]. Quantification by qPCR on genomic DNA or qRT-PCR on total RNA indicated that the deletion of exons 9 and 10 reached 95 % in follicular B cells (FoB) from *CD21-Cre Hells^{F/F}* mice and was almost complete in *Mb1-Cre Hells^{F/F}* FoB (Supplementary Fig. 2b, c). Even though mutant B cells do

express an mRNA with an in-frame deletion of exons 9 and 10 (Supplementary Fig. 2d, e), the corresponding truncated HELLS protein is clearly unstable as it could not be detected by western-blot with a polyclonal antibody that targets the C-terminal domain encoded by exons downstream the deletion (Supplementary Fig. 2f). We conclude that HELLS loss-of-function in B cells is most likely complete in these models, that are hence referred to as $Mb1Hells^{KO}$ and $CD21Hells^{KO}$ thereafter.

Total B cell numbers were normal in peripheral blood and in spleen of $Mb1Hells^{KO}$ mice (Fig. 2a), and splenic transitional T1 (B220⁺IgM⁺CD93⁺CD23ⁿᵉᵍ), T2 (B220⁺IgM⁺CD93⁺CD23⁺), FoB (B220⁺CD93ⁿᵉᵍCD21⁺CD23⁺) and marginal-zone (MZB, B220⁺CD21ʰⁱᵍʰCD23ˡᵒ) B-cell populations were found in equal proportions in control and $Mb1Hells^{KO}$ mice (Fig. 2b, c). In the bone marrow, pre-pro B/pro B (B220ˡᵒʷIgMⁿᵉᵍCD93⁺CD43⁺CD25ⁿᵉᵍ = Hardy's Fr. A and B), pre-B (B220ˡᵒʷIgMⁿᵉᵍCD93⁺CD43ⁿᵉᵍCD25⁺ = Fr. C' and D) and immature B cells (B220ˡᵒʷIgM⁺CD93⁺ = Fr. E) did not show any overt anomaly in $Mb1Hells^{KO}$ mice (Supplementary Fig. 2g). *Hells* expression in the B-lineage is thus dispensable for the development and the homeostasis of splenic B cells.

### *Hells* expression in B cells is mandatory for efficient TD humoral immune responses and establishment of high-affinity MBCs

As a prominent feature of ICF patients is hypogammaglobulinemia, we measured the immunoglobulin titers in the serum of non-immunized animals by ELISA. Compared to control littermates, $Mb1Hells^{KO}$ mice showed a significant decrease of basal IgG1, IgG2b, and IgG3 titers (60%, 46% and 77% reduction, respectively), but not of IgM and IgA (Fig. 2d). Several IgG sub-classes were similarly reduced in $CD21Hells^{KO}$ mice (Supplementary Fig. 3a), with the notable exception of IgG3. As IgG3 is associated with T-independent antibody responses, this difference between the two mouse models may be a consequence of the lack of Cre-mediated deletion in CD21ˡᵒʷ B-1a cells in *CD21-Cre* mice[51].

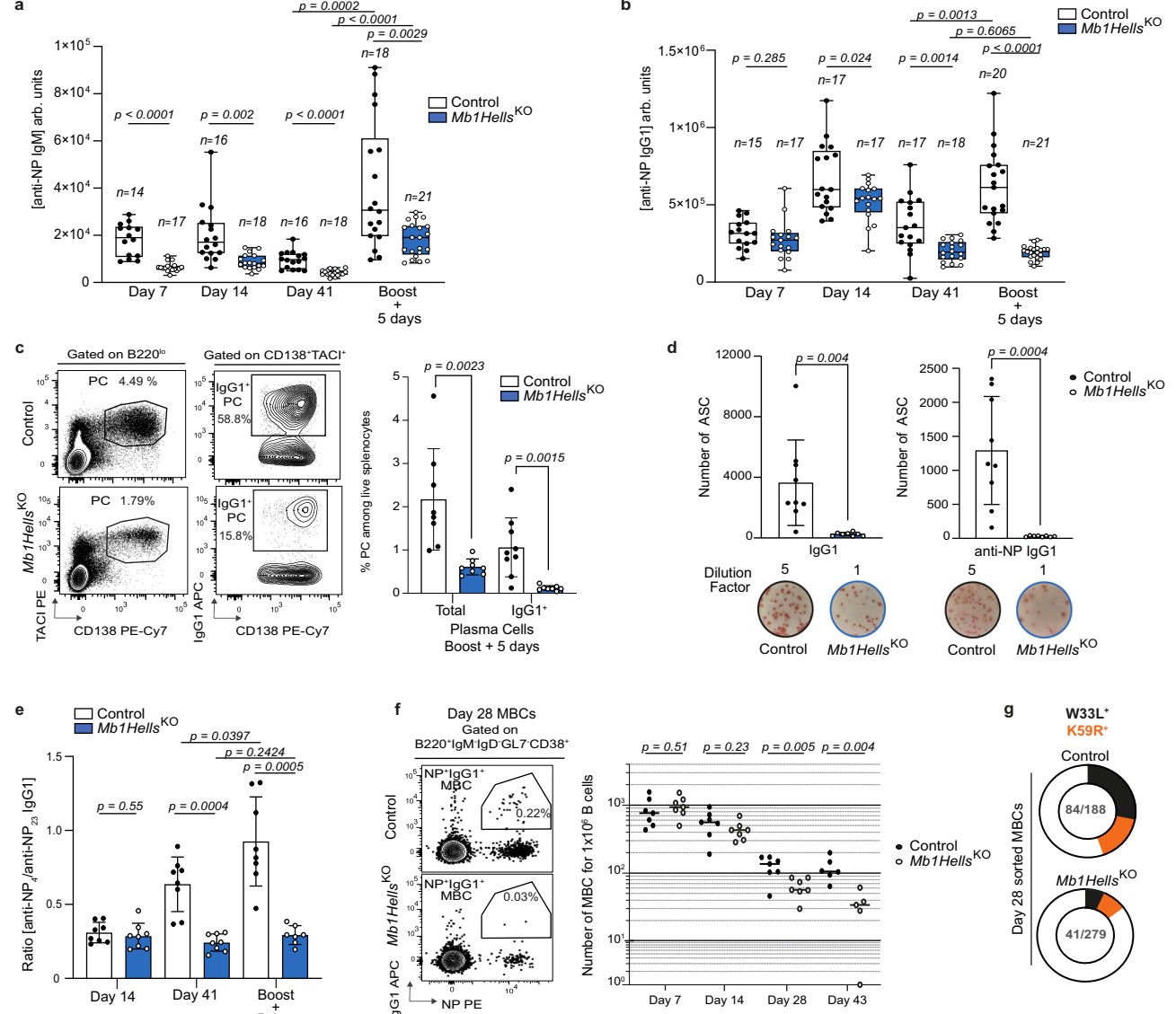

**Fig. 3 | Hells expression in B cells is mandatory for efficient TD humoral immune responses and establishment of high-affinity MBCs. a, b** Mice were immunized i.p. with alum-adsorbed NP-CGG and boosted 6 weeks later. Serum was collected at day 7, 14, and 41 after the primary immunization, and 5 days after the boost. NP-specific IgM (**a**), and IgG1 (**b**) from $Mb1Hells^{KO}$ and control animals (*n* for numbers of mice used is indicated in figures) were quantified by ELISA (arb.units, arbitrary units). **c** Representative Flow Cytometry (FC) plots of splenic IgG1-secreting cells 5 days after antigenic boost, and quantification of these cells (*n* = 8 for $Mb1Hells^{KO}$, *n* = 9 for controls). **d** Enumeration by ELISPOT assay of splenic IgG1-ASC and NP-specific IgG1-ASC, 5 days after antigenic boost. Each well is representative of a triplicate (*n* = 8 for $Mb1Hells^{KO}$, *n* = 9 for controls). **e** $NP_4$-BSA and $NP_{23}$-BSA-binding IgG1 were quantified by ELISA in the serum of the animals described in

(**b**) (*n* = 8 for each genotype) and the $NP_4/NP_{23}$ ratio was calculated. **f** Representative FC plots of splenic NP-specific IgG1⁺ MBC 28 days after immunization with NP-CGG, and quantification of NP-specific IgG1⁺ MBCs per million B220⁺ B cells in the spleen, 7, 14, 28 and 43 days after immunization (*n* = 7 for each genotype). **g** Proportion of VH186.2 BCR sequences from single-cell sorted NP-specific IgG1 MBCs at day 28 post NP-CGG immunization that bear W33L or K59R substitutions in control (*n* = 7) and $Mb1Hells^{KO}$ (*n* = 8) animals. The numbers of sequences (W33L or K59R/total) are indicated at the center of the pies. Experiments (**a**) to (**f**) were done twice, (**g**) was done three times, and data were pooled. Bar charts and error bars represent the mean ± SD. Unpaired two-tailed Welch's *t*-test were performed for (**a**), (**b**) and (**e**), and unpaired two-tailed *t*-test were performed for (**c**), (**d**) and (**f**). Source data are provided in Source Data File.

We conclude that the absence of HELLS in murine B cells also leads to hypogammaglobulinemia, though milder than that of some ICF patients[25].

We then assessed the capacity of the mutant mice to mount humoral immune responses against the T-dependent (TD) antigen 4-hydroxy-3-nitrophenyl (NP) hapten conjugated to Chicken Gamma Globulin (CGG). We injected control and mutant mice intraperitoneally (i.p.) with NP-CGG adsorbed on alum, and measured by ELISA the relative serum titers of NP-specific antibodies after primary and secondary immunizations; we focused on IgM and IgG1 isotypes as alum elicits $T_H2$-biased polarization. The humoral response to NP-CGG was

altered in multiple ways in $Mb1Hells^{KO}$ mice. Firstly, the primary IgM response was severely diminished in the mutant animals at all time points studied (day 7, 14, and 41, Fig. 3a). In stark contrast, the primary IgG1 response was initially equal in $Mb1Hells^{KO}$ and controls at day 7, and was only slightly decreased at the peak of the response around day 14 (Fig. 3b). However, at distance from the primary immunization, anti-IgG1 titers declined prematurely, leading to a 57% reduction relative to controls at day 41 that may reflect a reduced population of LLPCs. Secondly, despite rather normal primary IgG1 response, the secondary IgG1 response 41 days later was almost ablated in the absence of HELLS: 5 days after a boost, anti-NP IgG1 titers of $Mb1Hells^{KO}$ were only

one third as high as those of controls (Fig. 3b). In agreement with this observation, at the same time point, the mutant mice displayed a 90% decrease of splenic antibody-secreting cells (ASC), as assessed by flow cytometry (percentage of CD138$^+$TACI$^+$IgG1$^+$, Fig. 3c) or by ELISPOT (IgG1 secreting cells and NP-specific IgG1 secreting cells, Fig. 3d). $CD21Hells^{KO}$ presented with an identical lack of IgG1 anamnestic response (Supplementary Fig. 3b, c).

In parallel, we monitored the maturation of antibody affinity by measuring the relative amounts of IgG1 antibodies able to bind NP$_4$-BSA (high affinity anti-NP antibodies only) or NP$_{23}$-BSA (all anti-NP antibodies) in ELISA assays. While the ratio of NP$_4$-binding/NP$_{23}$-binding IgG1 increased over time after the primary and the secondary immunization in control animals, it remained stable in $Mb1Hells^{KO}$ even after a boost (Fig. 3e), which reflects a lack of secretion of high affinity antibodies.

We then enumerated by flow cytometry the NP-specific IgG1$^+$ MBCs which support the recall response by differentiating rapidly into ASCs upon antigen re-encounter. At distance from the primary immunization (day 28 and 43), the frequency of NP-specific IgG1 MBCs, identified as B220$^+$IgM$^{neg}$IgD$^{neg}$IgG1$^+$CD38$^+$GL7$^{neg}$, was reduced by two-thirds in both $Mb1Hells^{KO}$ and $CD21Hells^{KO}$ (Fig. 3f and Supplementary Fig. 3d). We single-cell sorted NP-specific IgG1$^+$ MBCs at day 28 post-immunization, and sequenced the rearranged V$_H$186.2 segment which dominates the antibody response to this hapten. The fraction of MBCs that bear the W33L or K59R substitutions known to confer increased affinity for NP[52,53] was severely reduced in $Mb1Hells^{KO}$ (Fig. 3g). The MBC compartment from $Mb1Hells^{KO}$ mice thus shows both quantitative and qualitative defects that account for their defective IgG1 anamnestic response.

## Germinal center B cells devoid of HELLS form and proliferate normally, but collapse prematurely

HELLS requirement for antibody affinity maturation and for recall responses points to a critical role during the germinal center reaction. We followed by flow cytometry the GC reaction in the spleen after a primary immunization with NP-CGG. The proportion of GC B cells, identified as B220$^+$CD95$^{high}$GL7$^{high}$ (Fig. 4a) or B220$^+$CD38$^{low}$GL7$^{high}$ (Supplementary Fig. 4a), was comparable in mutant and control mice after 7 days, at the onset of GCs. Despite a normal GC initiation, their decay was much faster in the absence of HELLS: at day 10, the percentage of GC B cells in $Mb1Hells^{KO}$ was already diminished by 40% compared to controls, and this reduction worsened at day 14 (two-fold reduction) and day 28, at which point GC B cells had almost disappeared in the mutant strain while they persisted over 40 days in littermate controls (Fig. 4a and Supplementary Fig. 4a). Very similar results were observed in $CD21Hells^{KO}$ mice, with a slightly milder phenotype (Supplementary Fig. 4b). Chronic GC response to commensal flora in intestinal Peyer's patches was also sharply lowered in non-immunized $Mb1Hells^{KO}$ (Fig. 4b). Apart from this premature collapse, GC physiology, as assessed by flow cytometry, was apparently preserved in the mutant animals. Two-dimensional analysis of cell cycle phases after in vivo labeling with 5-ethynyl-2′-deoxyuridine (EdU) did not reveal any blockade of $Mb1Hells^{KO}$ GC B cells in a specific phase, neither at day 10 (Supplementary Fig. 4c), nor at day 14 (Fig. 4c). In line with this normal proliferation, these GCs displayed a regular architecture with normal proportions of light-zone centrocytes (CXCR4$^{low}$CD86$^{high}$) and dark-zone centroblasts (CXCR4$^{high}$CD86$^{low}$) (Fig. 4d). Moreover, the mutant GC B cells which remained 14 days after immunization exhibited an equivalent somatic hypermutation load in the J$_H$4 intron to that of control cells (Fig. 4e), and the frequency of affinity-increasing mutations in the rearranged V$_H$186.2 sequence was not reduced either (Fig. 4f). These features indicate that the few GC B cells remaining in late GCs in $Mb1Hells^{KO}$ mice survived as many rounds of cell division as control cells and are not mainly newcomers that recently entered the GC reaction. The

absence of high-affinity MBCs suggests that most MBCs were formed early in young germinal centers while late GCs failed to yield long-lived MBCs.

All together, these results show that HELLS is dispensable for GC B cell differentiation, proliferation and affinity maturation of the BCR, but its loss-of-function causes an accelerated decay of GC B cells, which notably hampers the establishment of high-affinity MBCs and PCs.

## $Hells^{KO}$ GC B cells undergo global DNA hypomethylation associated with de-repression of retrotransposons, satellite DNA, and some germline genes

To better characterize the molecular phenotype caused by the lack of HELLS and the mechanism that leads to GC B cell disappearance, we first tracked variations of DNA methylation during the course of GC reaction, in accordance with its central role in ICF syndrome. We profiled cytosine methylation genome-wide using enzymatic-methyl sequencing (EM-seq) of naive, day 7 and day 14 GC B cells from control and $Mb1Hells^{KO}$ mice.

As previously shown[54], control GC B cells underwent a slight progressive genome-wide loss of DNA methylation (Fig. 5a). Hells-deficient naive B cells showed a massive loss of CpG methylation which increased further during the GC reaction, leaving <10% of methylated CpGs after 14 days (Fig. 5a and Supplementary Fig. 5a–c). This concerned all types of repeated sequences whose methylation was already known to depend on HELLS[29,32,33,55], namely satellite sequences, Short Interspersed Nuclear Elements (SINE), Long Interspersed Nuclear Elements (LINE) and Long Terminal Repeat (LTR) retrotransposons (Fig. 5a). Targeted bisulfite sequencing of DNA from day 10 GC B cells confirmed this severe hypomethylation in $Mb1Hells^{KO}$ (Supplementary Fig. 5d). Single-copy sequences were affected as well, with a few notable exceptions among CpG-dense promoters, some of which exhibited a slower kinetics of demethylation or even full resistance (Fig. 5b and Supplementary Fig. 5e). These data show that HELLS activity is necessary for DNA methylation maintenance during the proliferation steps of B-cell development and B-cell activation, except for some loci that are protected from demethylation through a HELLS-independent pathway.

We then profiled gene expression by bulk RNA sequencing of naive, day 7 and day 14 GC B cells sorted from control and $Mb1Hells^{KO}$ mice to identify potential transcriptomic disturbances and correlate them with CpG methylation. While almost all types of retrotransposons remained silent in naive B cells, some specific families of endogenous retroviruses (ERV) became massively de-repressed in day 7 GC B cells; this phenomenon was further amplified in late GC B cells (Fig. 5c, d). In particular, the evolutionary recent ERV types Intracisternal A-particle (IAP, subfamilies Ez and Ey)[56] showed a strong and progressive reactivation, in good agreement with previous observations on demethylated embryonic stem cells, embryos and neurons[57].

Quantification by qRT-PCR on total RNA confirmed the release from silencing of IAPs and also showed a 6-fold overexpression of minor satellite transcripts (Fig. 5e). Regarding single-copy loci, only few genes displayed significant transcriptional changes (adjusted-BH $p$ value < 0.05), most of them being upregulated (120 and 257 upregulated genes with FC ≥1,5 at day 7 and 14, respectively, vs. 37 and 41 downregulated genes with FC ≤ 0.66 at day 7 and 14, respectively). Among those, we identified a set of de-repressed single-copy loci, i.e., genes whose transcription was detectable in mutant cells while it remained null in control cells (Fig. 5f), which enlarged as the GC reaction progressed (Supplementary Fig. 5f).

The analysis of the top-50 de-repressed genes showed that nearly half of them harbour an IAPEz element in an intron or at a distance <10 kb upstream or downstream the gene (Fig. 5g); in the former case, only exons downstream the IAPEz-int copy were significantly

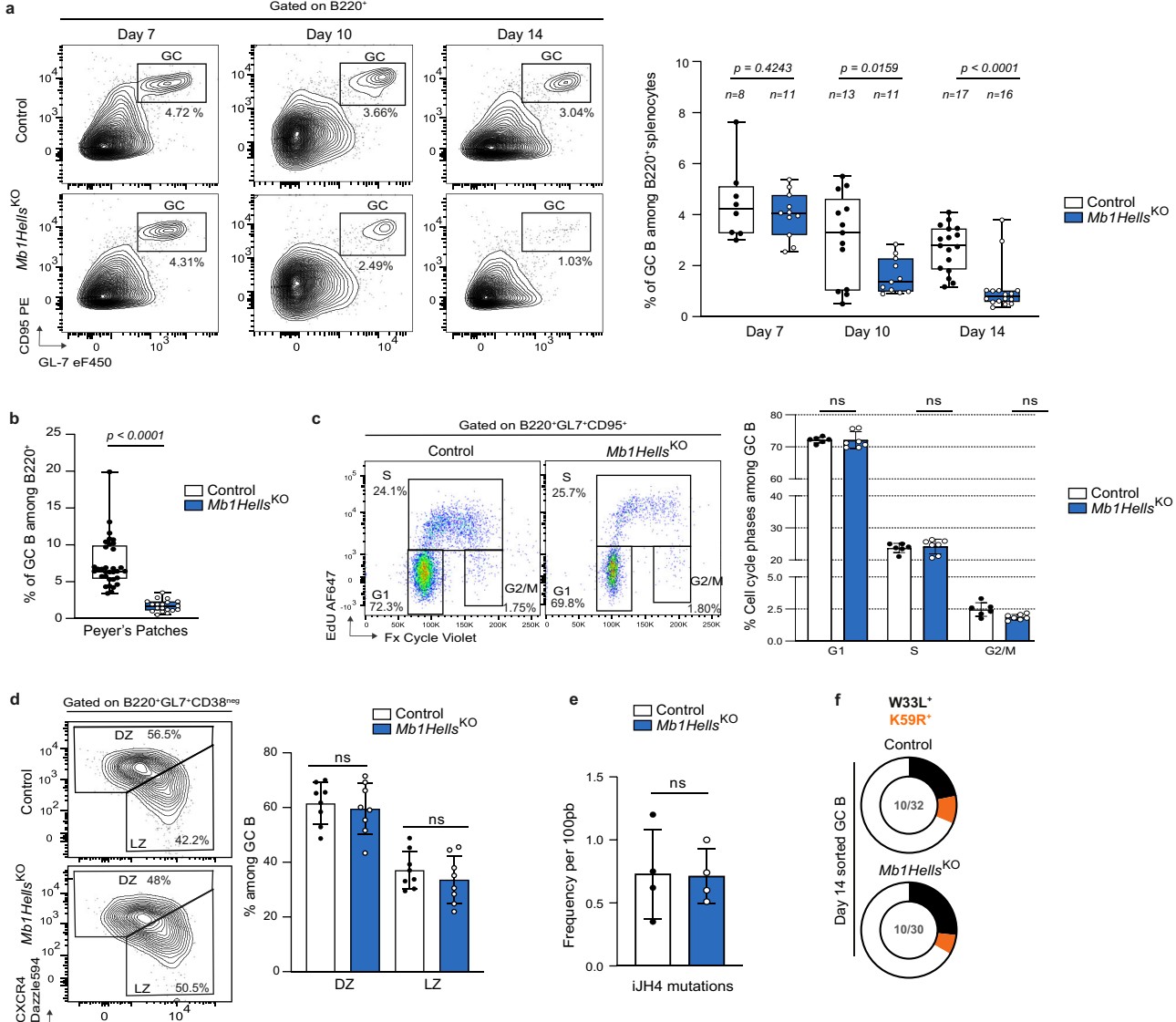

**Fig. 4 | GC B cells devoid of *Hells* form and proliferate normally but collapse prematurely. a** GC kinetics in the spleen after immunization. Representative Flow cytometry (FC) plots and quantification of GC B cells in *Mb1Hells^KO* and control animals, 7-, 10- and 14-days post NP-CGG immunization (*n* for numbers of mice used is indicated in the figure). **b** Percentage of GC B cells among living Peyer's patches B cells measured by FC (*n* = 19 for *Mb1Hells^KO* and *n* = 33 for control mice, of which 4 were *Mb1^Cre/WT^Hells^F/WT*, pooled from multiple independent experiments). **c** 2D-cell cycle analysis of day 14 GC B cells after in vivo EdU labeling. Representative FC plots (left) and quantification of cell cycle phases. Data shown for *n* = 7 animals of each genotype (2 controls were *Mb1^Cre/WT^Hells^F/WT*). **d** Distribution of splenic GC B cells between Light Zone (LZ) and Dark Zone (DZ) 14 days after immunization. Representative FC plots for LZ and DZ B cells (left), and quantification of their percentages within each zone (right, *n* = 8 for each genotype). **e** Somatic hypermutation frequency in GC B cells. Bar charts show the frequency of intron JH4 mutations per 100 pb in GC B cells 14 days after immunization with NP-CGG. *n* = 4 animals of each genotype. **f** Proportion of VH186.2 BCR sequences from sorted GC B cells at day 14 post NP-CGG immunization that bear W33L or K59R substitutions in control and *Mb1Hells^KO* animals. The numbers of sequences (W33L or K59R/total) pooled from 4 mice of each genotype are indicated at the center of the pies. Experiments were done twice; data were pooled for (**a**), (**c**), (**d**), (**e**), and (**f**), and one representative experiment was shown for (**c**). Bar charts and error bars represent the mean ± SD. Unpaired two-tailed *t*-test were performed for (**a**), (**b**), (**c**), (**d**) and (**e**); ns: non-significant. Source data are provided in Source Data File.

transcribed (Supplementary Fig. 5g). In contrast, among the top-50 upregulated genes (i.e., whose expression is already detectable in control GC B cells), only 3 include an IAPEz element in the close vicinity, which suggests that IAPEz de-repression contributes to the production of aberrant gene-derived transcripts in GC B cells deficient for HELLS. In addition, as previously observed in ICF patients' leukocytes[58], several de-repressed genes devoid of IAPEz-int correspond to genes well-known for their expression in immunoprivileged tissues such as brain and testis, and previously shown to be controlled by DNA methylation, such as *Tex13b*[59], *Rhox2a, 2h* and *5*[60], *Sohlh2*[61] or *Tuba3*[62]

(Fig. 5g). In agreement, when we unbiasedly looked at the expression of genes with CpG-dense promoters and not expressed in wild-type GC B cells, we observed the appearance of some level of transcription only for demethylated promoters and never for that resistant to demethylation (Supplementary Fig. 5h). The observed transcriptional de-repression (Fig. 5f, g and Supplementary Fig. 5h) mostly consists in mild transcription—probably due to the absence of relevant transcription factors in the B lymphoid lineage—but it is still noticeable since it shows that removal of the epigenetic barrier allows for potential transcriptional activation[63].

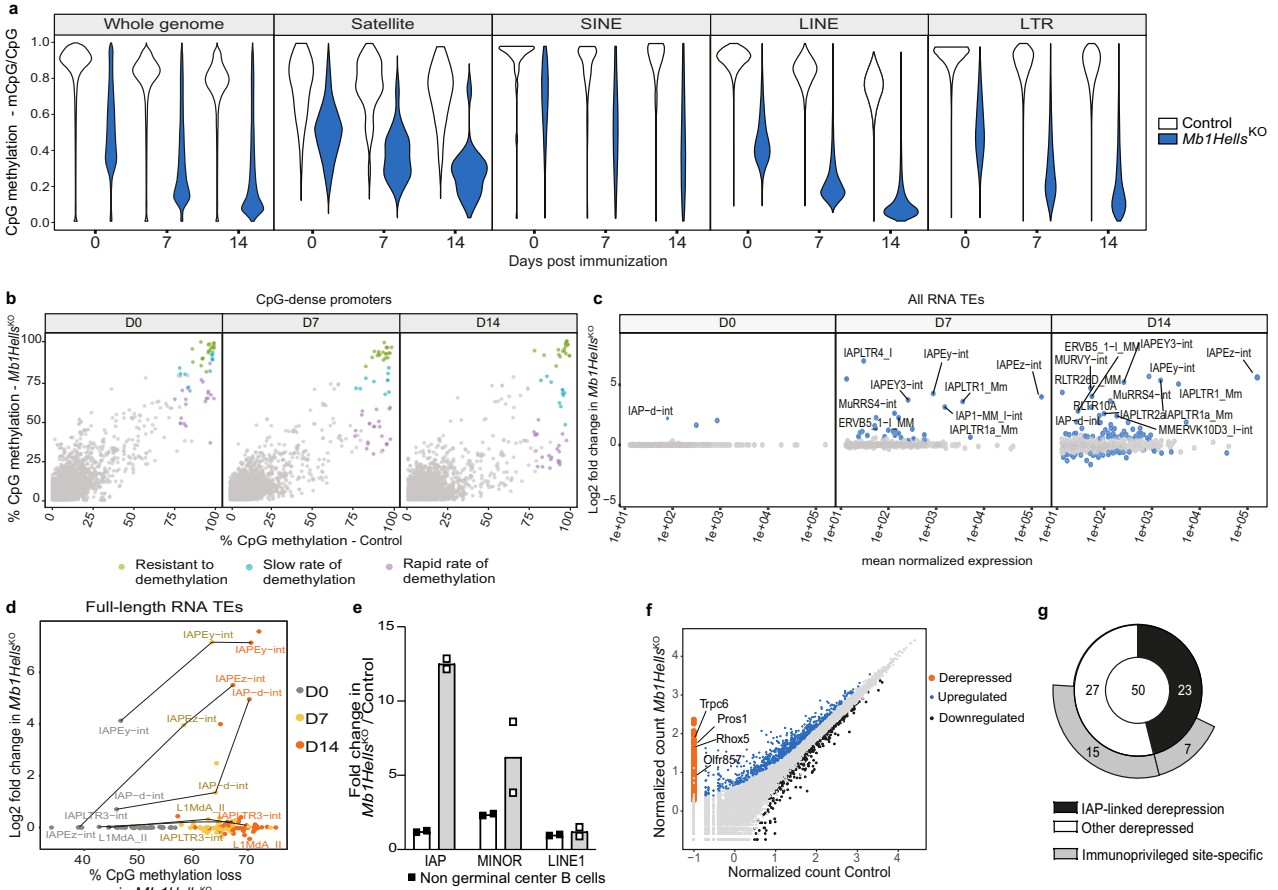

**Fig. 5 | *Hells*^KO GC B cells undergo deep hypomethylation and derepression of repeated non-coding sequences and germline genes. a** DNA methylation levels of naïve (day 0), day 7 and 14 GC B cells sorted from *Mb1Hells*^KO and control animals. The distribution of mCpG/CpG ratio is shown by violin plots for 200 CpGs genomic tiles (left) and different classes of repetitive elements (right). The average of two biological replicates is shown. **b** Correlation plots of CpG methylation levels at CpG-dense promoters (% CpG > (0.25)²*100) between *Mb1Hells*^KO and control animals. Promoters with WT methylation >75% at all time points were clustered based on the kinetics of methylation changes in *Mb1Hells*^KO using *k*-means clustering. **c** MA plots displaying the differential expression of RNA Transposable Elements (TEs) between *Mb1Hells*^KO and control samples at the three time points. *X* and *Y* axis show the average of normalized count values and the shrunken log2 fold changes, respectively. Colored dots have adjusted *p*-value < 0.05, calculated by a two-tailed Wald

test. **d** Correlation between CpG methylation loss in *Mb1Hells*^KO samples and reactivation kinetics (DESeq2 shrunken log2 fold change) for full-length RNA TEs (LINE1 > 5 kb, IAP > 6 kb and MMERVK10C > 4.5 kb). **e** Relative amounts of IAPEz, minor satellite and LINE-1 transcripts in naïve and day 10 GC B cells from *Mb1Hells*^KO mice compared to controls (*n* = 2 for each genotype). Transcripts were quantified by RT-qPCR and normalized to GAPDH. **f** Differential gene expression between *Mb1Hells*^KO and control day 14 GC B cells measured by bulk RNAseq (*n* = 4 for each genotype). Upregulated (fold change>1.5, *p* < 0.05) and downregulated genes (fold change < 0.65, *p* < 0.05) are represented in blue and black, respectively. Orange dots represent derepressed genes, whose transcripts are detected only in *Mb1Hells*^KO cells. **g** Classification of the top-50 derepressed genes, according to the presence of an IAPEz copy, or to expression restricted to immunoprivileged tissues. Source data are provided in Source Data File.

## HELLS loss-of-function does not induce excess cell death in GC B cells

The abnormal accumulation of transcripts arising from repeated non-coding sequences and ERVs may be toxic to GC B cells, which could explain their premature decay. Such transcripts may induce viral mimicry, which could lead to autocrine type-I interferon secretion and subsequent cell death as previously described in models with genetic or pharmacological inhibition of the DNA methylation maintenance machinery[64–67]. Surprisingly, neither IFN-β nor IFN-α was detected in mutant GC B cells by RNA-seq, and GSEA analysis of upregulated genes showed no evidence of enrichment of type-I IFN targets (Supplementary Fig 8a). Alternatively, de-repression of ERVs and germline-specific genes in *Hells*^KO GC B (Fig. 5c, g and Supplementary Fig. 5g) cell may be a source of neo-epitopes that would heighten their immunogenicity and subsequent targeting by cytotoxic CD8⁺ T cells, mimicking an anti-tumoral response[68–71]. However, mutant GC B cells show no sign of elevated caspase activity as assessed by cytometry with anti-activated caspase 3 antibody, or with a pan-caspase fluorogenic substrate

(Supplementary Fig. 6a, b), which rules out excess apoptosis in these cells. Also, depletion of cytotoxic CD8⁺ T cells by injection of anti-CD8α antibody after the onset of GC reaction failed to preserve them from accelerated decay (Supplementary Fig. 6c, d).

HELLS was recently shown to prevent ferroptosis, an iron-dependent form of nonapoptotic cell death, through the activation of several metabolic genes including *Fads2*[72] whose expression was indeed reduced to one-third in day 14 *Mb1Hells*^KO GC B cells (Supplementary Fig. 6e). However, the labeling with BODIPY-C11 probe, a fluorescent reporter for lipid peroxidation characteristic of ferroptotic cells[73], was similar in control and mutant GC B cells (Supplementary Fig. 6f). All in all, these results do not support the hypothesis that HELLS loss-of-function increases GC B cell death.

## Accelerated acquisition of pre-memory B cell markers in the absence of HELLS

As the premature disappearance of HELLS-deficient GC B cells could not be accounted for by either proliferation blockade or increased cell

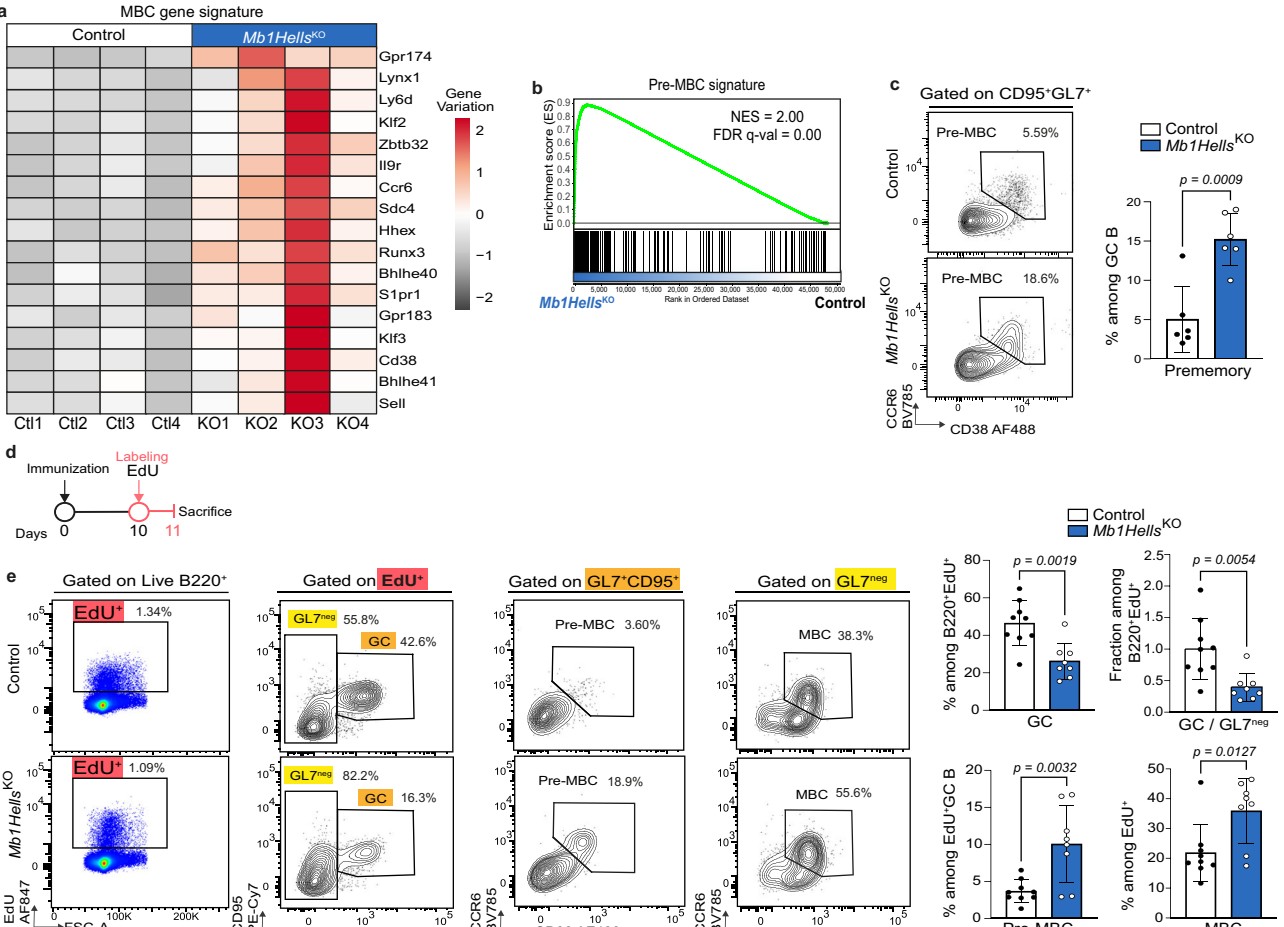

**Fig. 6 | Accelerated acquisition of pre-MBC markers in the absence of Hells.** **a** Heatmap representation of scaled counts of selected genes in control and *Mb1Hells^KO* mice. **b** GSEA showing enrichment of a pre-MBC signature in *Mb1Hells^KO* vs. control day 14 GC B cells. The pre-MBC signature consists in the top 200 upregulated genes in pre-MBC vs. GC B cells in GSE89897. **c** Representative flow cytometry (FC) plots showing the CD38⁺CCR6⁺ pre-MBC population within GC B cells (B220⁺CD95⁺GL7⁺), and quantification of the pre-MBC subset in *Mb1Hells^KO* (n = 6) and control (n = 6) mice. **d** Experimental setup for in vivo EdU pulse labeling of GC B cells and follow-up by flow cytometry. **e** Representative FC plots showing the phenotype of EdU positive live B cells 24 h after the in vivo pulse in one *Mb1Hells^KO* and one control mice. Frequencies of GC (CD95⁺GL7⁺), pre-MBC (CD95⁺GL7⁺CD38⁺CCR6⁺) and MBC (GL7⁻CCR6⁺CD38⁺) were quantified in n = 8 *Mb1Hells^KO* and n = 9 control mice. Experiments **c** and **e** were done twice, and data were pooled. Bar charts and error bars represent the mean ± SD. Unpaired two-tailed *t*-test were performed. Source data are provided in Source Data File.

death, we assumed that it could reflect a premature exit from the GC. RNA-seq analysis of day 14 GC B cells indeed revealed the upregulation in mutant cells of many markers normally repressed in centroblasts and centrocytes, and re-expressed in MBCs, such as genes coding for transcription factors (*Hhex, Bhlhe40, Bhlhe41, Runx3, Zbtb32, Klf2,* and *Klf3*), or adhesion molecules and surface receptors (*Cd38, Ccr6, Il9r, Sdc4, S1pr1, Ly6d,* and *Lynx1*) (Fig. 6a). Using a public dataset[74], we defined a pre-MBC transcriptional signature that includes the top-200 genes upregulated during pre-MBC differentiation from GC B cells (Source Data – Fig. 6) and ran a GSEA analysis which confirmed a very significant enrichment of this signature in day 14 *Mb1Hells^KO* cells relative to control GC B cells (Fig. 6b). Examination by flow cytometry indicated that this enrichment resulted from a threefold increase in the percentage of a CCR6⁺CD38⁺GL7^int^CD95^int^ population (15% of GC B cells in the mutants vs. 5% in controls, Fig. 6c), that likely corresponds to pre-MBCs[75]. Importantly, 24 h after an in vivo EdU pulse (Fig. 6d), which mainly labels GC B cells, the percentage of GL7⁺CD95⁺EdU⁺ B cells that acquired a pre-memory B cell phenotype was tripled in mutant mice, while the proportion of EdU⁺ B cells that had lost GL7 and CD95 markers was significantly increased in the same mice (Fig. 6e). This observation suggests that HELLS-deficient GC B cells are less able to maintain their phenotype and part of them engage prematurely in

the pre-MBC differentiation pathway. In addition, these GC B cells are not exceedingly replaced by new immigrants as late GC B cells in mutant mice harbor normal somatic hypermutation level and their hypomethylation keeps increasing between days 7 and 14 (Figs. 4e, f and 5a).

## Hells^KO GC B cells hyperactivate an mTORC1/ATF4-dependent metabolic pathway typical of ASC differentiation

As the premature GC exit through MBC differentiation only partially explains GC decay, we performed additional GSEA analysis of the genes abnormally upregulated in day 14 *Mb1Hells^KO* GC B cells, which uncovered an enrichment of three overlapping signatures: mTORC1 pathway, Unfolded Protein Response, and ATF4 targets (Fig. 7a). Actually, only a fraction of the genes of each signature was significantly overexpressed in the absence of HELLS; for example, genes involved in the sterol biosynthesis pathway were on the contrary repressed in day 14 *Hells^KO* GC B cells, even though they are part of the mTORC1 pathway (Fig. 7a). A set of 33 genes recurrently contributed to the three signatures, coding for factors involved in amino acid metabolism (*Asns, Chac1, Bcat1, Psph, Phgdh, Psat1, Pycr1, Alh18a1,* and *Sesn2*) and transport across the membrane (*Scl6a9, Slc1a4, Slc3a2,* and *Slc7a5*), aminoacyl-tRNA synthesis (*Tars, Cars, Gars, Mars, Iars, Nars, Yars,* and

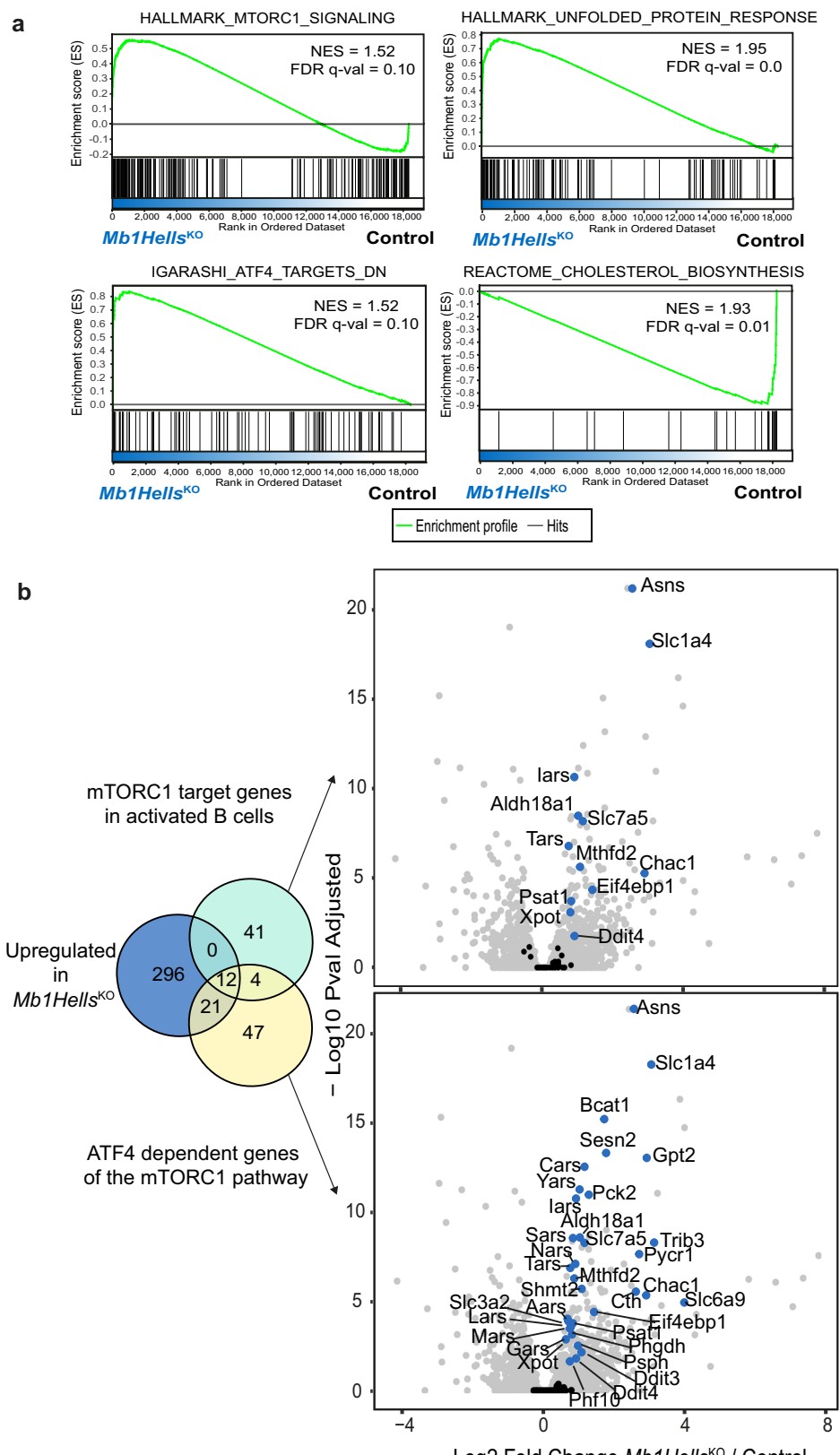

**Fig. 7 | *Mb1Hells^KO* GC B cells overexpress an mTORC1/ATF4-dependent metabolic signature. a** GSEA showing enrichment of signatures related to mTORC1 activation, UPR, ATF4 targets and cholesterol biosynthesis in *Mb1Hells^KO* day 14 GC B cells. NES normalized enrichment score, FDR false discovery rate. **b** Venn Diagram of upregulated genes in day-14 *Mb1Hells^KO* GC B cells compared to mTORC1 targets upregulated in activated B cells (GSE141423) and ATF4-dependent mTORC1 targets (GSE158605). Genes commonly upregulated in day-14 *Mb1Hells^KO* GC B cells and in GSE141423 and/or GSE158605 are represented in blue on the volcano plot (gray and blue = *p* < 0.05, calculated by a two-tailed test). Source data are provided in Source Data File.

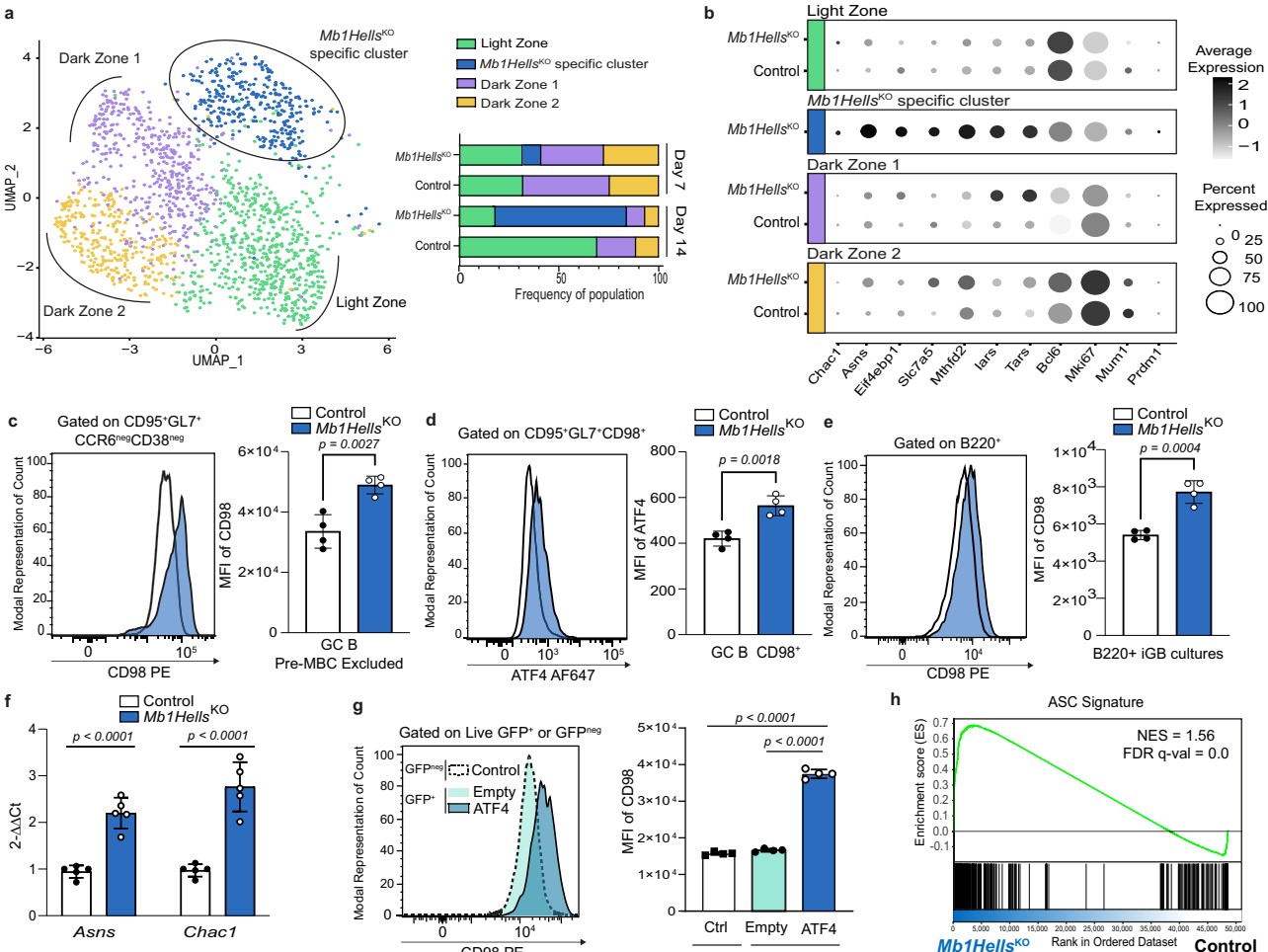

**Fig. 8 | *Hells^KO* GC B cells overexpress an mTORC1/ATF4-dependent metabolic pathway upregulated during ASC differentiation. a** Single-cell transcriptional analysis of day 7 and 14 GC B cells. UMAP displaying 429 day-14 GC B, 466 day-7 GC B *Mb1Hells^KO*, 414 day-14 GC B, and 448 day-7 GC B control cells, colored by shared nearest neighbor clusters collected. Bar charts show the frequencies of GC B cells within the 4 clusters. *n* = 2 mice for each genotype and timepoint; one of the two control mice for day-7 GC B cells was *Mb1^Cre/WT^Hells^F/WT^*. **b** Selected gene expression for the 4 different clusters shown in (**a**), presented in dot plot scaled on normalized UMI counts. **c** Representative CD98 staining of CD95⁺GL7⁺CCR6^neg^CD38^neg^ GC B cells and quantification of CD98 geometric MFI (Mean Fluorescent Intensity). **d** Representative intracellular ATF4 staining within CD95⁺GL7⁺CCR6^neg^ GC B cells and quantification of ATF4 geometric MFI (Mean Fluorescent Intensity). **e** Representative CD98 staining of B220⁺ day-8 iGB cell cultures and quantification of CD98 geometric MFI (Mean Fluorescent Intensity). **f** Expression of *Asns* and *Chac1* in sorted B220⁺CD138^neg^ day-8 iGB cells measured by RT-qPCR after normalization to *Ubc*. **g** Representative CD98 staining of GFP⁺ and GFP^neg^ day-8 iGB cell cultures after transduction of empty or ATF4 coding retroviral vectors, and quantification of CD98 geometric MFI (Mean Fluorescent Intensity). **h** GSEA showing enrichment of ASC signature in *Mb1Hells^KO* vs. control day 14 GC B cells. The ASC signature consists in the top 200 genes upregulated in GSE60927. NES normalized enrichment score, FDR false discovery rate. *n* = 4 for each genotype in (**c**), (**d**), (**e**), (**f**) and (**h**); experiments were done twice and one representative experiment is shown. *n* = 5 for (**g**); experiment was done twice and data were pooled. Bar charts and error bars represent the mean ± SD. Unpaired two-tailed *t*-test were performed for (**c**) to (**g**). Source data are provided in Source Data File.

*Lars*), one-carbon metabolism and purine synthesis (*Mthfd2*, *Shmt2*, and *Cth*). ATF4 is the central transcription factor of the Integrated Stress Response (ISR) triggered by viral infection, amino acid deprivation, heme deprivation or endoplasmic reticulum stress[76]. Each of these stresses activates a specific kinase that phosphorylates eIF2α, which once phosphorylated represses general mRNA translation while licensing ATF4 translation[76]. Interestingly, ATF4 translation is also enhanced in a p-eIF2α independent manner, through phosphorylation of 4E-BP proteins by mTORC1[77]. ATF4 was thus recently identified as one of the transcriptional effectors downstream mTORC1 activation[78] and most interestingly, the transcription targets described in this study encompass our whole set of genes (Fig. 7b) that we designate as mTORC1/ATF4 signature thereafter.

To determine whether this signature is restricted to a small population or widely overexpressed in *Mb1Hells^KO* GC B cells, we conducted single-cell RNA sequencing (scRNA-seq) on GC B cells from mutant and control mice, 7 and 14 days after immunization and excluded from the sorting procedure the pre-memory B cell population already identified. To this end, GC B cells were single-cell sorted using a B220⁺CD95^hi^GL7^hi^ gate and an average of 440 cells were studied for each genotype at each timepoint. While three main clusters corresponding to light zone and two dark zone subsets (Supplementary Fig. 7a) were present in almost equal proportions in day 7 *Mb1Hells^KO* and control GCs, an additional cluster emerged specifically in the mutant mice, and expanded dramatically at day 14, including more than 70% of *Mb1Hells^KO* GC B cells at that time (Fig. 8a). This specific cluster stands out from the others by its expression of the genes that appeared de-repressed because of the presence of an IAPEz element within the gene, or in its vicinity (Supplementary Fig S7b). Moreover, this same cluster shows the highest expression of the mTORC1/ATF4 targets presented above (Fig. 8b), demonstrating that this signature is upregulated in a large majority of mutant day 14 GC B cells. Moreover,

cell surface expression of the CD98 heavy chain increased homogeneously by 60% in cells lacking HELLS (Fig. 8c), a figure that matches the upregulation of *Slc3a2* (1.69 fold-increase) and *Slc7a5* (2.22 fold-increase) mRNAs which code for CD98 heavy and light chains, respectively. In agreement with this conclusion, intracellular flow cytometry showed a slight, but reproducible and uniform 1.4 fold overexpression of ATF4 in CD98^high *Mb1Hells^KO* cells (Fig. 8d), and CD98 was used as surrogate marker of this pathway in the following experiments.

To confirm that ATF4 controls the expression of this mTORC1/ATF4 signature in B cells, we turned to an in vitro culture system which enables differentiation of naive B cells into highly proliferating GC-like B cells (iGB), and later on into ASCs[47]. After 4 days of culture with IL-4 on feeder cells which express CD40L and BAFF, followed by 4 days in the presence of IL-21, the resulting *Hells^KO* iGB cells exhibited a uniform 1.5-fold overexpression of surface CD98 relative to control iGB21 cells (Fig. 8e), as well as a two- to three-fold overexpression of *Asns* and *Chac1* mRNA (Fig. 8f). Having validated the hyperactivation of our signature in these in vitro conditions, we then transduced WT B cells with a retroviral vector that contained the ATF4 coding sequence without its inhibitory 5' translation leader sequence[79]. Cells that overexpressed ATF4 expressed two-times more CD98 than non-transduced cells, or cells transduced with an empty vector (Fig. 8g).

Taken together, these results indicate that HELLS loss-of-function in B cells leads to ATF4 overexpression in activated B cells, which turns on a specific metabolic program. This does not seem to be a consequence of Integrated Stress Response (ISR), as in vivo treatment of *Mb1Hells^KO* mice with the ISR inhibitor ISRIB at doses previously shown to modulate brain function[80] neither restrained premature GC decay, nor CD98 overexpression (Supplementary Fig. 7c). Moreover, only a minor fraction of ATF4 targets usually upregulated during ISR was indeed upregulated in *Hells*-deficient GC B cells, and as explained above, these targets had been shown to be specifically induced after mTORC1 activation which only slightly raises ATF4 translation[78]. We thus favor the hypothesis that this signature reflects a subtle mTORC1 hyperactivation in GC B cells that cannot be detected by intracellular flow cytometry for pS6 (Supplementary Fig. 7d) as described elsewhere[81]. About half of the genes that compose this signature are part of the mTORC1-dependent metabolic program induced by B cell activation (Fig. 7b), which prepares these cells to subsequent differentiation into ASC and massive antibody secretion[82]. The expression profile of these genes inferred from Immgen database confirmed that most of them are particularly highly expressed in splenic plasmablasts and plasma cells (Supplementary Fig. 7e). More generally, GSEA analysis on the genes upregulated in *Hells^KO* GC B cells with a plasma cell signature extracted from the literature[83] showed a significant enrichment of this signature (Fig. 8h). Since mutant day 14 GC B cells retained a relatively high *Bcl6* expression level and scarcely expressed the ASC-determining transcription factors *Mum1*/IRF4 and *Prdm1*/BLIMP1 (Fig. 8b), we propose that the hyperactivation of this mTORC1/ATF4 pathway reflects a very early commitment to ASC lineage that would occur prematurely in the absence of HELLS.

### Pharmacological inhibition of DNMT1 phenocopies the impact of HELLS loss-of-function on GC kinetics

HELLS was recently shown to repress some gene transcription independently of its role in DNA methylation, through deposition of macroH2A histone variants[84,85]. In an attempt to determine the contribution of the loss of DNA methylation maintenance to the phenotype caused by *Hells* knockout in B cells, we immunized WT mice with NP-CGG, and seven days later, once GCs were established, we treated them for seven consecutive days with a DNMT1-specific inhibitor (Fig. 9a). We chose GSK3685032 because it does not inhibit DNMT3A or 3B activity, and does not induce DNA breaks, contrary to decitabine[86]. Moreover, this compound has proven to have a good

in vivo tolerability since it did not show any overt toxicity in mice after a 28-day treatment[86]. GSK3685032 treatment caused a 5-fold reduction in the percentage of splenic GC B cells relative to injection of the vehicle (Fig. 9b) and the residual cells displayed significant hypomethylation of LINE-1 repeats as expected (18.2% of unmethylated CpG, vs. 7.9% for the vehicle, $p = 0.0014$, Fig. 9c). This massive loss of GC B cells was accompanied by a fourfold increase in the percentage of CCR6^+CD38^+ pre-MBCs among CD95^+GL7^+ GC B cells (Fig. 9d), and a 1.4-fold increase in the surface expression of CD98 in CCR6^-CD38^-CD95^+GL7^+ GC B cells (Fig. 9e). We extended this observation to a 3-day in vitro culture using LPS, IL-2 and IL-5 to induce ASC differentiation, supplemented with GSK3685032 at concentrations ranging from 100 nM to 1 μM. Cell-trace violet dilution assay revealed that GSK3685032 impeded cell proliferation in a dose-dependent manner (Fig. 9f). At concentrations above 300 nM, the DNMT1 inhibitor enhanced CD98 expression and increased the percentage of B220^lowCD138^high ASCs, and both effects were particularly marked for cells that had undergone at least 3 rounds of division (Fig. 9g, h). As a whole, these results show that the phenotype conferred by pharmacological inhibition of DNA methylation maintenance is reminiscent of that observed for *Mb1Hells^KO* mice, and that the premature decay of GC B cells in the absence of HELLS may be a direct consequence of an accelerated exit from the GC stage caused by DNA hypomethylation.

## Discussion

Our study shows that murine B cells require a functional HELLS enzyme for efficient TD-antibody responses, which are short-lived, of poor affinity and lack anamnestic capacity in its absence. Since the early IgG1 response as well as GC initiation are preserved in the mutant mice, we conclude that this deficiency is not a consequence of a general defect in B-cell activation, proliferation or ASC differentiation, and it is likely independent of the otherwise proposed function of HELLS and ICF pathway in class-switch recombination[43,87]. Our data rather point to a defect in GC B cell maintenance that would impede the generation of LLPCs and high affinity MBCs, which arise with distinct kinetics during the GC reaction[88–90].

We favor the hypothesis that the critical function of HELLS for GC B cell persistence is DNA methylation maintenance, since its deletion causes progressive and global loss of CpG methylation in GC B cells, and treatment with a DNMT1 inhibitor similarly disrupts established GCs. As the lack of HELLS in murine B cells phenocopies important features of the ICF immunodeficiency, such as hypogammaglobulinemia and reduction of the MBC compartment, we propose that this immunodeficiency is at least partly consecutive to a failure to maintain DNA methylation in B cells. In accordance with this model, a mutation in the DNMT1 co-factor UHRF1 was recently diagnosed in a single ICF patient, though with a mild immune phenotype[91]. We note however that the hypomethylation in *Hells^KO* GC B cells is drastic compared to that observed in blood leukocytes of ICF4 patients[91,92], which may reflect a requirement of CDCA7/HELLS complex for DNA methylation maintenance of wider genomic regions in GC B cells, beyond late-replicating heterochromatic regions[37,40,41]. We favor the hypothesis that the progressive demethylation observed in the absence of HELLS is likely caused by passive demethylation consecutive to inefficient DNA methylation maintenance, rather than by active demethylation through Tet-mediated hydroxymethylation, in agreement with a previous study on several HELLS-deficient cell lines[36]. The peak of expression of *Hells* during S phase (ref. 93 and our scRNA-seq data) thus fits with its requirement for DNA methylation maintenance in actively cycling GC B cells. The reason why HELLS deficiency causes such a severe hypomethylation in GC B cells therefore remains an open question, but it is noteworthy that maintenance of cytosine methylation is a particularly demanding process for centroblasts, which proliferate extremely fast with shortened S-phase[94], and undergo active DNA demethylation consecutive to DNA repair triggered by AID-

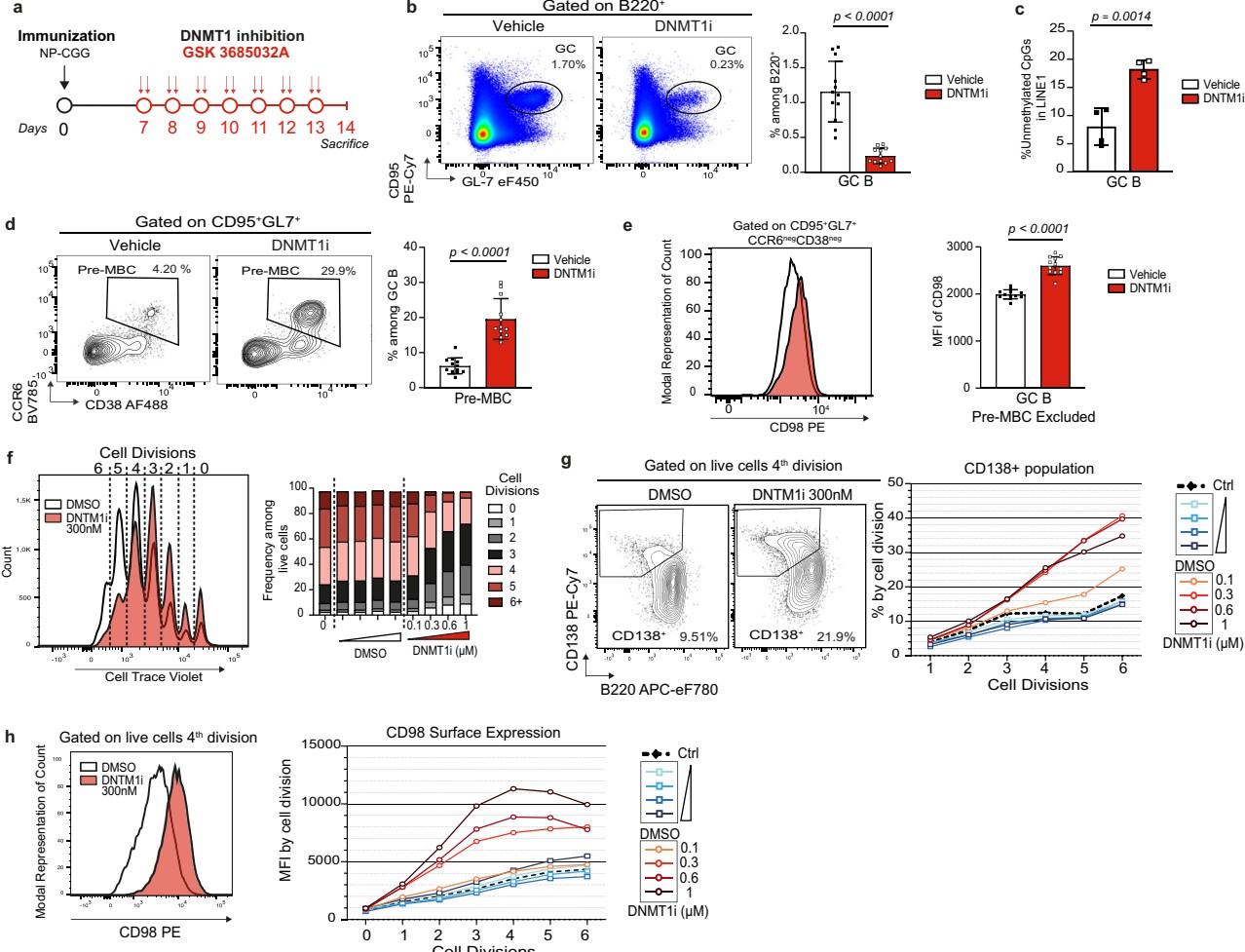

**Fig. 9 | Pharmacological inhibition of DNMT1 phenocopies the impact of HELLS loss-of-function on GC kinetics. a** Experimental setup for DNMT1 inhibition in vivo. **b** Representative flow cytometry (FC) plots showing splenic GC B cell population after treatment with vehicle or DNMT1 inhibitor GSK3685032, and quantification of GC B percentage. **c** Percentage of unmethylated CpGs inferred from bisulfite sequencing of 16–20 LINE1 copies amplified from sorted GC B cells after 5 days of treatment with DNMTi or vehicle. **d** Representative FC plots showing the pre-MBC population within GC B cells after treatment with vehicle or GSK3685032, and quantification of pre-MBC percentage. **e** Representative CD98 staining within $CD95^+GL7^+CCR6^{neg}CD38^{neg}$ GC B cells after treatment with vehicle or GSK3685032, and quantification of CD98 geometric MFI (Mean Fluorescent Intensity). **f** Representative Cell Trace Violet staining after 3 days of in vitro culture with LPS + IL-2 + IL-5, in the presence of 300 nM GSK3685032 or DMSO.

Distribution of living cells according to the number of cell divisions, for increasing concentrations of GSK3685032 or DMSO. **g** Representative FC plots showing ASC (CD138⁺B220⁺) differentiation after 3 days of in vitro culture as in (**e**). Frequencies of ASCs as a function of cell division number, for increasing concentrations of GSK3685032 or DMSO ($n = 2$ independent cultures). **h** Representative CD98 staining of cells that underwent 4 divisions after 3 days of in vitro cultures as in (**e**) and quantification of CD98 geometric MFI (Mean Fluorescent Intensity) as a function of cell division number and of GSK3685032 or DMSO concentrations ($n = 2$ independent cultures). All experiments were done twice; data were pooled for (**b**) to (**e**); one representative experiment is shown for (**f**), (**g**), (**h**). $n = 12$ mice for each treatment in (**a**), (**b**), (**d**). $n = 4$ mice for each treatment were used in (**c**). Bar charts and error bars represent the mean ± SD. Unpaired two-tailed $t$-test were performed for (**b**), (**c**), (**d**), (**e**). Source data are provided in Source Data File.

mediated deamination[54,95]: small differences in DNA methylation landscape are thus most likely accentuated during the GC reaction.

As the gene that codes for de novo DNA methyltransferase DNMT3B is most frequently mutated in ICF patients[23,24], an impairment of proper DNA methylation landscape prior to B-cell activation could also abrogate the GC reaction. The ICF phenotype may thus arise as a consequence of the failure to maintain a global DNA methylation state, that would converge for all ICF patients on common loci during the GC reaction[40,41,92].

Alternatively, as DNMT3B is upregulated in human GC B cells (Fig. 1a), it may participate to DNA methylation maintenance in these cells, as previously reported in embryonic cells or intestinal epithelium[96–100]. This idea is supported by the finding that DNMT3B acts as a co-activator of other DNMTs independently of its catalytic activity[101]. In murine GC B cells, which express low levels of *Dnmt3b* but

abundantly express *Dnmt1*[102] (Fig. 1b, c and Supplementary Fig. 1a), the maintenance of CpG methylation marks is likely to rely exclusively on DNMT1 and UHRF1, and DNMT3B loss-of-function in murine B cells does not impede GC persistence[103]. The importance of HELLS-dependent DNA methylation maintenance in GC B cells does not exclude that other HELLS functions, such as transcription repression through macroH2A deposition, DNA hydroxymethylation or DNA repair through NHEJ, may also contribute to B-cell physiology.

How could failure of DNA methylation maintenance affect GC B cell persistence? The massive de-repression of repeated sequences consecutive to genome-wide hypomethylation was expected to cause a major cellular stress: accumulation of dsRNA or other non-coding RNA had been described to induce PERK-dependent Integrated Stress Response and a type-I interferon response through viral mimicry[64–67], and reactivation of endogenous retrovirus was shown to induce

deleterious UPR response in mouse pro-B cells[104] and premature senescence in human cells[105]. Surprisingly, no induction of a type-I IFN response was detected in *Hells*[KO] GC B cells, which suggests that cytosolic sensing of nucleic acids is physiologically mitigated in these cells (Supplementary Fig. 8b, c), possibly to prevent toxic type-I IFN secretion arising after spontaneous loss of DNA methylation[54]. It is noteworthy that genome-wide hypomethylation affected neither GC B cell proliferation nor survival. This is in stark contrast with the cell-cycle arrest caused by UHRF1 loss-of-function in the same cells[106], but this phenotypic difference may be explained by a deeper hypomethylation in the *Uhrf1*[KO] model. Instead, hypomethylated GC B cells displayed subtle transcriptomic perturbations, with inappropriate expression of genes typical of either MBC or ASC precursors, which we interpret as an accelerated kinetics of GC exit responsible for the premature GC decay. While the loss of DNA methylation had already been shown to play a positive role in ASC differentiation[107,108], our results point to a more general role of DNA methylation maintenance in preventing premature exit from the GC. Conversely, the moderate demethylation that naturally occurs after each round of cell division could shape the evolution of GC output in aging GCs; this hypothesis is supported by our observation that part of the upregulated genes in day 14 *Hells*[KO] GC B cells are similarly upregulated in day 21 WT GC B cells (Supplementary Fig. 8d[109]). Whether the defective early IgM response observed in *Mb1Hells*[KO] mice similarly reflects an expedited extrafollicular reaction with precocious class switching, or an independent function of HELLS remains an open question.

Since this accelerated GC kinetics does not result in increased antibody titers or MBC numbers, but instead gives rise to transient antibody secretion and almost absent secondary responses, we conclude that the excessively hypomethylated MBC and ASC precursors are hardly able to differentiate into fully functional long-lived cells. Only reduced numbers of low-affinity MBCs and ASCs would form in these mice, during the early steps of extrafollicular and GC reactions when hypomethylation is still moderate. Although supported by in vivo EdU labeling experiments, the hypothesis of a premature exit in the form of abortive MBC and ASC precursors awaits further validation through genetic fate-mapping. This impairment may result from an incomplete engagement of the appropriate transcriptional programs, such as the mTORC1-dependent metabolic program required for ASC differentiation. *Hells*[KO] GC B cells upregulate many genes involved in amino acid transport and metabolism as a result of ATF4 over-expression, but underexpress SREBP2 targets which code for factors involved in sterol metabolism, known to favor plasma cell differentiation[110]. While there is no clear explanation for this apparent dissociation of mTORC1-dependent transcriptional pathways, our results point to a previously unappreciated role for ATF4 in ASC physiology. In addition, the apparent re-expression of RIG-I in MBCs (Supplementary Fig. 8c) may restore their susceptibility to the accumulation of non-coding RNAs transcribed from de-repressed repeated sequences and compromise their survival.

Details about how hypomethylation hampers the repression of post-GC transcriptional programs in HELLS deficient mice are still missing. Despite severe and genome-wide loss of CpG methylation, the number of deregulated genes in the absence of HELLS remains rather low and, apart for genes near an IAPEz element, there is generally no correlation between the methylation status of the promoter and the transcription level, as expected[111]. This does not preclude that a small number of genes may be directly upregulated consecutively to the loss of CpG methylation of binding sites for methylation-sensitive transcription factors[57,112] in promoter or enhancer regions, but the coverage of our genome-wide methyl-seq analysis was not enough to identify candidate genes. We note, however, that the deregulation of only a few genes may have broader transcriptomic consequences: abnormal expression of a gene that modifies the concentration of some amino acids could on its own subtly hyperactivate mTORC1,

resulting in enhanced ATF4 translation and further upregulation of other ATF4 targets. In this regard, the Asparagine synthetase was recently proposed to promote on its own an effector phenotype of CD8+ T cells[113].

The critical role of DNA methylation maintenance in GC persistence and humoral immunity suggests it could be an interesting pharmacological target to treat some antibody-mediated auto-immune diseases or GC-derived B-cell neoplasms. The pan-DNMT inhibitor 5-Azacytidine was recently shown to ameliorate auto-immune arthritis in an experimental mouse model, by inhibiting germinal center reaction and attenuating IgG1 secretion[114]. Our results indicate that the DNMT1-selective inhibitor GSK3685032, or small compounds able to disrupt HELLS-CDCA7 interaction may offer safer alternatives with fewer deleterious side effects. Reciprocally, finding ways to sustain DNA methylation maintenance after immunization may enhance vaccination schemes against pathogens such as HIV, for which neutralization requires prominently prolonged affinity maturation.

In whole, our study sheds light on an alternative mechanism of GC exit, in which affinity maturation is not the sole driver of GC B cell fate.

## Methods

### Inclusion and ethics

The research described in this article complies with all relevant ethical regulations. All animal studies were ethically reviewed by Ethics Committee of Paris Descartes University (CEEA34, authorizations no. APAFiS #6691-2016062816418985, and APAFiS #31365-2021030513547842) and Anses/EnvA/Upec Ethics Committee (C2EA-16, authorization no. APAFiS #9951-2017051611079789), validated by the French Ministry of Research, and carried out in accordance with the European Directive 2010/63/EEC. The study on pediatric splenic samples was conducted in compliance with the Declaration of Helsinki. Parents from pediatric patients provided written informed consent before the collection of splenic samples. Spleen samples were conserved and prepared with the authorization of the Comité de Protection des Personnes Ile de France II (DC-2008-448).

### Mouse strains, immunization and treatments

Mice with LoxP sites flanking *Hells* exons 9 and 10 (*Hells*[F] allele[43]) were crossed with *CD21-Cre* (*Tg(Cr2-cre)3Cgn*[50];) or *Mb1-Cre* (*Cd79atm1(cre)Reth*[49]) strains. CD21-Cre and Mb1-Cre mice were kindly provided by Patrick Revy (Institut Imagine, Paris, France) and Simon Fillatreau (INEM, Paris, France), respectively. The resulting offspring were inter-crossed and maintained on a mixed C57BL/6 J:129 genetic background at the SFR Necker animal facility (Paris), in individual ventilated cages with enrichment, under specific and opportunistic pathogen-free (SOPF) conditions. Animals were fed standard chow diet (Teklad Global 2918, 18% protein, irradiated) ad libitum, and kept under ambient temperature (21–22 °C) and 50–60% humidity, with 12 h–12 h on-off light cycle. 6-weeks-old to 6-months-old mice of both sexes were used for all experiments. We did not conduct post hoc sex-based analysis as the study had not been specifically designed for this.

Genotyping of mice was performed by PCR on DNA prepared from biopsies by overnight incubation at 56 °C with 20 μg of Proteinase K (Roche) diluted in a buffer composed of 5 mM EDTA, 200 mM NaCl, 100 mM Tris-HCl pH 8.5 and 0.2% SDS, followed by inactivation at 95 °C for 10 min. Genotyping primer sequences and cycling conditions are described in Supplementary Table 1.

Mice over 6 weeks were immunized intra-peritoneally (i.p.) with 100 μg of NP-CGG (4-hydroxy-3-nitrophenylacetyl-Chicken Gamma Globulin, Biosearch Technologies) dissolved in PBS and adsorbed on *Alhydrogel*® aluminium hydroxide following manufacturer's instructions (Invivogen). To study recall response, mice were reimmunized i.p. with NP-CGG in alum 6 weeks later. Blood was taken at the facial vein using 5 mm Golden Rod lancets (Braintree Scientific), and serum

was separated after blood clotting by two serial centrifugations (300 g, 5 min at 4 °C).

Treatment with GSK3685032A DNMT1 inhibitor (DNMTi): 6-week-old male and female C57BL6/J SOPF mice purchased from Charles River (France) were injected subcutaneously twice daily with 30 mg.kg$^{-1}$ of GSK3685032A dissolved in Captisol (Cydex) from day 7 to day 13 after immunization with NP-CGG and euthanized by cervical dislocation one day later. Control animals received the vehicle only.

Treatment with anti-CD8α: mice were injected i.p. with 200 μg of anti-CD8α or isotype control at day 7 after immunization with NP-CGG and euthanized by cervical dislocation at day 14. Control animals received the vehicle only.

Treatment with ISRIB. ISRIB solution was made by dissolving 5 mg ISRIB (Tocris, #5284) in 2.5 mL DMSO (PanReac AppliChem, 191954.1611). The solution was gently heated in a 40 °C water bath and vortexed every 30 s until the solution became clear. Next, 1 mL of Tween 80 (Sigma Aldrich, P8074) was added, the solution was gently heated in a 40 °C water bath and vortexed every 30 s until the solution became clear. Next, 10 mL of polyethylene glycol 400 (PEG400) (PanReac AppliChem, 142436.1611) solution was added, gently heated in a 40 °C water bath and vortexed every 30 s until the solution became clear. Finally, 36.5 mL of 5% dextrose (Hospira, RL-3040) was added. The solution was kept at room temperature throughout the experiment. Mice were injected i.p. with ISRIB solution (2.5 mg.kg$^{-1}$) or vehicle, each day from day 8 to day 13 after NP-CGG immunization.

Treatment with EdU: mice were injected i.p. with 1 mg.kg$^{-1}$ EdU (InVitrogen) 10 days or 14 days after NP-CGG immunization.

### Pediatric human B cells
Spleen samples were obtained from one 4.5-years-old male child, and one 6-years-old female child, undergoing a splenectomy for a non-immunological disease-related reason (sickle cell disease). Spleen mononuclear cells were obtained, respectively from mechanically disrupted spleen samples after a density gradient centrifugation on Ficoll-Paque PLUS (GE Healthcare).

### Western blot
$4 \times 10^6$ B cells activated with LPS and IL-4 for 3 days were resuspended in 300 μL Laemmli 1× buffer with β-mercaptoethanol and sonicated with a Bioruptor device (10 × 30 s ON / 30 s OFF). After centrifugation (5 min, 14,000 × $g$, +4 °C), 20 μL were denatured 3 min at 95 °C, charged on a 4–20% TGX gel (Bio-Rad), separated in denaturing conditions, and transferred onto a 0.45 μm nitrocellulose membrane (Bio-Rad) that was subsequently blocked with Intercept kit (Li-Cor). HELLS and α-tubulin were detected respectively with a rabbit IgG polyclonal antibody (11955-1-AP, ProteinTech; dilution 1:2000 in Li-Cor diluent) and a mouse monoclonal antibody (T6199, Sigma; dilution 1:5000 in Li-Cor diluent), and revealed with Li-Cor IRDye® 680RD Goat anti-Mouse IgG and IRDye® 800CW Goat anti-Rabbit Secondary Antibodies diluted 1:15,000 in Li-Cor diluent, using an Odyssey® CLx Infrared Imaging device.

### Flow cytometry analysis and cell sorting
Single cell suspensions were prepared from mouse spleen, bone marrow, and Peyer's Patches in cold HBSS medium with Ca$^{2+}$ and Mg$^{2+}$ (Gibco) supplemented with 10% FBS (Fetal Bovine Serum, Gibco), and filtered through a 40 μm nylon mesh (BD Biosciences). Red blood cells from spleen and bone marrow were lysed by hypotonic shock thanks to a 30 s incubation at room temperature (RT) in erylysis buffer (0.747% NH$_4$Cl, 1.7 mM Tris-HCl pH 7.2), followed by immediate addition of cold HBSS-10% FBS and centrifugation at 300 g for 5 min at 4 °C. Cells were washed in HBSS-10% FBS and counted automatically on Cellometer (Nexcelom Bioscience) after exclusion of dead cells with trypan blue.

Surface staining of unfixed cells with fluorochrome-conjugated antibodies were performed on $1–10 \times 10^6$ cells in 100 μL of PBA (PBS without Ca$^{2+}$Mg$^{2+}$ (Gibco), supplemented with 0.5% BSA) for 20 min on ice in the dark, except for iGB cell cultures that were stained at RT. Cells were subsequently washed with PBA and finally resuspended in cold PBS-2% FBS-2 mM EDTA. Dead cells were labeled with 1 μM Sytox Blue (Molecular Probes) or with a 5 min incubation with 1% 7-AAD (BioLegend), just prior FACS analysis. For NP-specific staining, dead cells were stained for 30 min with Live Dead Aqua (0.1% in PBS, Molecular Probes), according to manufacturer's instructions. Cells were subsequently washed with PBA, and stained for 45 min with the mix of antibodies together with NP-PE (2.5 μg.mL$^{-1}$; N-5070-1, Biosearch Technologies). For intracellular stainings, cells were permeabilized and fixed with BD Cytofix/Cytoperm (BD Biosciences) following manufacturer's guidelines, before intracellular staining for IgG1, ATF4 or pS6. Cell cycle analysis was performed after cell surface staining using Click-iT™ Plus EdU AF647 (Thermo CF10634) on day 10 or on day 14 post NP-CGG immunization, per manufacturer's instructions. After Click-iT reaction, DNA staining was carried out for 30 min using FxCycle™ Violet stain (Thermo F10347) in the permeabilization reagent. All data were acquired on BD FACS Canto II or LSRFortessa devices with DIVA software (BD Biosciences) and analyzed with FlowJo software (version 10.8.1, Treestar Inc.). B cell populations were purified either through cell sorting with a BD FACSAria III, or through negative selection on magnetic beads with EasySep™ Mouse B Cell Isolation Kit (Stemcell Technologies).

For detection of apoptotic cells, total splenocytes were incubated 30 min in R15 medium with pan-caspase inhibitor CaspGLOW (Invitrogen, 88-7003-42), then labeled with surface markers. Alternatively, total splenocytes were incubated 90 min in R15 medium, then labeled with surface markers, fixed and stained with anti-activated caspase 3 antibody.

For measurement of lipid peroxidation in GC B cells, total splenocytes were incubated for 30 min at 37 °C in RPMI medium containing 2 μM of BODIPY-C11 (Invitrogen, D3861), before quantification of the signal in the FITC channel. A positive control for lipid peroxidization was obtained by incubating splenocytes with 2 or 4 μM RSL3 (Selleckchem, S8155) during 20 h at 37 °C, prior to staining with BODIPY-C11.

Antibodies are listed in Supplementary Table 2.

### ELISA
F96 MaxiSorp immunoplates (Nunc) were coated during 1 h at 37 °C with 50 μL per well of a 10 μg.mL$^{-1}$ solution of goat anti-mouse Ig (#1010-01, SouthernBiotech), or of NP-BSA capture antigens of different valencies (NP4, 23 or 41-BSA, Biosearch Technologies) in coating buffer (carbonate-bicarbonate, pH 9.6), to detect total Ig sub-classes or NP-specific antibodies. Plates were then saturated overnight at 4 °C with 100 μL PBS-1% BSA per well, and incubated for 1 h at RT with 50 μL per well of serum samples serially diluted in PBS. Two duplicate dilution series were performed for each sample. After a last incubation with 100 μL of HRP-conjugated goat anti-mouse IgG1, IgM, IgG2b, IgG2c, IgG3 or IgA (SouthernBiotech, 5300-05B) diluted 2000 times in PBS-1% BSA- 0.05% Tween 20, each well was incubated 5–10 min at RT with 50 μL of KPL SureBlue TMB Microwell Peroxidase Substrate (SeraCare) and the reaction was stopped by addition of a 0.6 N H$_2$SO$_4$ solution. Five washes with PBS were performed between each step described above, and before the revelation step with TMB substrate. Optical density was measured at 450 nm with SkanIt Software 5.0 for Microplate Readers RE, and background signal measured at 620 nm was substracted. We calculated the basal seric Ig titer by using commercial unlabeled mouse Ig of each isotype (SouthernBiotech, 5300-01B). For NP-specific Ig titers, a reference pool of sera taken from control mice 7 days after i.p. immunization with NP-CGG (or NP-Ficoll,

for IgG3) was used as an internal control on every plate to calculate relative Ig titers.

## ELISPOT assays

Plates (MSIPS410, Millipore) were activated 1 min with 15 μL of 35% ethanol per well, and subsequently washed once with $H_2O$, and twice with PBS, before coating of each well for 2 h at 37 °C with 50 μL of a 10 μg.mL$^{-1}$ solution of goat anti-mouse Ig (#1010-01, SouthernBiotech), NP23-BSA (Biosearch Technologies) or Keyhole Limpet Hemocyanin (KLH) diluted in coating buffer (carbonate-bicarbonate, pH 9.6). After one wash with PBS, and two with PBST (PBS-0.05% Tween 20), plates were saturated with 200 μL of PBS-1% BSA during 2 h at RT. Plates were then washed two times with PBS, and serial dilution of cells were transferred in individual wells and incubated overnight at 37 °C, 5% $CO_2$ in R15 B-cell medium (RPMI 1640 + Glutamax, supplemented with 15% characterized fetal bovine serum (FBS), 55 μM β-mercaptoethanol, 20 mM HEPES (pH 7.0), 1 mM sodium pyruvate, non essential amino acids (1%), 100 units.mL$^{-1}$ penicillin/streptomycin). All components were from Gibco, except the serum (HyClone). The day after, plates were washed three times with PBS, and three times with PBS-T. 100 μL of secondary HRP-conjugated goat anti-mouse Ig antibodies (1:2000, SouthernBiotech, 5300-05B) diluted in PBS-1% BSA-0.05% Tween20 were added to each well for 1 h at RT. Finally, after three washes in PBS, and in PBST, spots were revealed by a 10 min incubation with 3-amino-9-ethylcarbazole (BD Biosciences) followed by full rinsing with tap water, and enumerated with an ELISPOT reader using the AID software (version 3.5: Autoimmun Diagnostika).

## PCR, quantitative PCR, and RT-PCR

For genotyping, PCR analysis was performed using corresponding primers. Reactions were performed with 0.33 mM of each dNTP, 10 pmol of each primer, 1.25 u GotaqG2 (Promega) and 0,5 μL DNA, and run on 2% agarose gels stained with 1/10,000 SybrSafe (Invitrogen).

DNA and RNA extraction were performed with DNEasy, AllPrep DNA/RNA and RNEasy kit (with on-column DNAse treatment) from Qiagen, following manufacturer's instructions. Quantitative PCR and RT-PCR were performed on a CFX96 Touch™ Real-Time PCR Detection System (Bio-Rad), using the following program: 10 min at 95 °C, and 40 cycles of 10 s at 95 °C and 1 min at 60 °C. Relative quantifications were calculated by the ΔΔCt method.

Quantification of the deletion of murine *Hells* gene for $CD21Hells^{KO}$ in genomic DNA was achieved with Taqman Universal PCR Master Mix, noAmpErase UNG (Applied Biosystems), using a copy number TaqMan assay for *Hells* exon 9 (Mm00501856) and a reference locus *Tfrc* (4458370).

Reverse transcription was done at 42 °C with the AffinityScript Multiple Temperature cDNA Synthesis Kit (Agilent) and random hexamers, according to the instructions of the provider. Control reactions (RT-) were conducted in the absence of reverse transcriptase, to monitor gDNA contamination.

A fragment of *Hells* cDNA was amplified on RNA from $Mb1Hells^{KO}$ or control GC B cells with primers located within exons 3 and 13, using Taq Platinum DNA polymerase High Fidelity (Invitrogen) and the following cycling conditions: 2 min 94 °C, 35 cycles (15 s 94 °C, 30 s 55 °C, 2 min 68 °C), 4 min 72 °C. PCR amplicons were run on agarose gels, purified with Nucleospin Gel Extraction kit from Macherey Nagel, and Sanger sequenced.

To quantify the relative amounts of transcripts from specific genes, we used a cDNA quantity corresponding to a total of 1000–4000 cells, and TaqMan assays with Taqman Universal PCR Master Mix, noAmpErase UNG (Applied Biosystems). We used the following assays for murine transcripts: *Dnmt3b*, Mm01240113; *Zbtb24*, Mm01190324; *Cdca7*, Mm00788027; *Hells*, Mm00468580, and for

human transcripts: *DNMT3B*, Hs00171876; *ZBTB24*, Hs00207703; *CDCA7*, HS00230589; *HELLS*, Hs00934778). Relative transcript expression was normalized to the expression of β2-microglobulin (Mm00437762 and Hs00984230).

The relative amounts of transcripts derived from centromeric minor satellites, IAPEz, LINE-1 and *Hells* were measured with iTaq Universal SYBR Green Supermix qRT-PCR (Bio-Rad) and custom primers. Data were normalized to the expression of *Gapdh*. Same conditions were applied for *Asns* and *Chac1*, except that *Ubc* was used as reference housekeeping gene.

Primers are listed in the Supplementary Table 1.

## Bisulfite-specific cloning-based Sanger sequencing

gDNA was extracted from cells using AllPrep DNA/RNA Micro and Mini kits (Qiagen) according to the manufacturer's instructions. Bisulfite treatment that converts unmethylated cytosines into uraciles was performed on 0.2–1 μg thanks to EpiTect Plus Bisulfite kit (Qiagen). Regions of interest were amplified by PCR from 0.5–50 ng of converted gDNA, with Platinum Taq DNA polymerase High Fidelity (Invitrogen) under the following conditions: 2 min at 94 °C; 35 cycles with 15 s at 94 °C, 30 s at 53 °C, 1 min at 68 °C; 10 min at 68 °C. PCR products were run on agarose gels stained with SybrSafe (Invitrogen), bands of the expected size were cut and purified using NucleoSpin Gel and PCR Clean-up kit (Macherey Nagel), and cloned with TOPO TA Cloning kit (Invitrogen) following manufacturer's instructions. Plasmids were purified thanks to the 96-Well Plasmid and BAC Preparation Kit (Millipore) and inserts were sequenced with BigDyeTerminator v3.1 kit (Applied Biosystems) and analyzed on an ABI PRISM 3130× Genetic Analyzer.

## Somatic hypermutation in JH4 intron, and detection of W33L and K59R mutations in VH186.2 sequences

Fourteen days after primary immunization, B220$^+$CD95$^+$GL7$^+$ splenocytes were sorted, and the JH4 intron flanking rearranged VH sequences was amplified with a mixture of five VH primers designed to amplify most mouse VH families and a downstream primer allowing the determination of 443 bp of noncoding sequences downstream of rearranged JH4 segments, as previously described[115]. Mutations were detected with use of CodonCode Aligner (CodonCode Corporation).

VH186.2 DNA sequencing was performed for bulk GC B cells sorted 14 days post NP-CGG immunization GC B cells, and single-cell MBCs sorted 28 days post NP-CGG immunization.

For bulk analysis, 5000 to 15 000 live GC B cells (B220 + CD95 + GL7+) were sorted directly in lysis buffer for RNA extraction with Qiagen RNEasy kit. Nested PCR was carried out to amplify VH186.2-JH2. Outer primers bulk1_VH186.2 For and bulk 1_VH186.2 Rev (as listed in Supplementary Table 1) were used for PCR1 and inner primers bulk2_VH186.2 For and bulk2_VH186.2 Rev were used for PCR2 following: 98 °C for 2 min, 25 cycles of 15 s 98 °C, 30 s 60 °C and 30 s 72 °C; and 2 min at 72 °C, using Phusion high fidelity DNA Polymerase (Thermo). Amplified products were cloned with TOPO-TA Cloning kit (InVitrogen), and Sanger sequenced.

For single cell analysis, MBCs (B220+IgM-IgD-IgG1+PNA-CD38 + NP+) were single-cell sorted in 96-well Biorad plates (HSP9641) containing 4 μL lysis solution (10 mM DTT, 8 u RNAsine (Promega) in 1× PBS and $H_2O$). cDNA synthesis was carried out using an adapted protocol from[116]. Briefly, cDNA synthesis was performed at 50 °C using Superscript III (Thermo 18080093) and Cγ1 ext oligos (see Supplementary Table 1). Two rounds of PCR (15 min at 95 °C; 50 cycles of 30 s 94 °C, 30 s 57 °C and 1 min 72 °C; 7 min at 72 °C) were performed using HotStart Taq DNA polymerase (Qiagen), first using sc_Cg1Ext and sc_VH186.2 oligos, and a second nested PCR using sc_g1Int and sc_VH186.2 oligos, as listed in Supplementary Table 1. Amplified products were purified using Sephadex and sequenced.

### In vitro B cell cultures and retroviral transduction

Murine ATF4 coding sequence without upstream ORFs was PCR-amplified from Mouse ATF4 (CHOP11/cATF)-WT vector, a gift from David Ron (Addgene plasmid # 21845; http://n2t.net/addgene:21845; RRID:Addgene_21845), and cloned into *Bgl II* and *Xho I* sites of the pMIG retroviral vector, a gift from William Hahn (Addgene plasmid # 9044; http://n2t.net/addgene:9044; RRID:Addgene_9044), to generate pMIG-ATF4 expression vector.

Nonreplicative ecotropic γ-retroviral particles were produced by transient lipofection (XtremeGene 9 DNA transfection reagent, Roche) of HEK293T cells (ATCC CRL-3216) grown in DMEM/F-12 supplemented with 10% FCS (Hyclone), with pMIG or pMIG-ATF4, pLTR-env, and CMV-Gag/Pol vectors (ratio 2:1:1). 48 h after transfection, viral supernatants were filtered through 0.45-μm Minisart High Flow filters (Sartorius) and concentrated on centrifugal filter units (100 K Amicon Ultra 15; EMD Millipore).

B cells were cultivated for iGB cultures following a previously described protocol[47]. Briefly, cells were cultivated on 40LB stromal layer for 4 days with 1 ng.mL$^{-1}$ mIL-4 (Peprotech). For iPCs generation, iGB cells were seeded back on 40LB stromal cells with 10 ng.mL$^{-1}$ mIL-21 (Peprotech) for 4 more days. 40LB cells were a kind gift from Pr D. Kitamura (Tokyo University of Science, Japan). Transduction was performed on 3-days old iGB cultures. Polybrene (8 μg.mL$^{-1}$, Sigma H9268) was mixed with $2 \times 10^6$ TU retroviral particles and B-cell culture medium and incubated on ice for 10 min. iGB medium was gently removed, replaced with Polybrene mix, centrifuged for 1h30 at 32 °C at $1100 \times g$ and incubated for 2 h at 37 °C. Half of the Polybrene Viral containing medium was replaced with new B-cell medium containing mIL-4 (1 ng.mL$^{-1}$). Cells were washed and reseeded the next day on fresh 40LB stromal cells and mIL-21 (10 ng.mL$^{-1}$) for 4 more days.

For in vitro treatment with DNMTi GSK3685032A, total splenic murine B cells were purified through negative selection with magnetic beads (STEMCELL Technologies), labeled with 2 μM Cell-Trace Violet (Molecular Probes) to track cell division number, and cultured in vitro in R15 medium in the presence of LPS (25 μg.mL$^{-1}$, Sigma L6143), IL-2 (20 ng.mL$^{-1}$) and IL-5 (5 ng.mL$^{-1}$) (cytokines from Peprotech). GSK3685032A or DMSO was added to the cultures at various concentrations, and cells were analyzed by flow cytometry after 3 days.

All cell cultures were conducted at 37 °C, under a 5% $CO_2$ water-saturated atmosphere. 293T and 40LB cell lines were not authenticated in our laboratory, but they displayed the expected morphology and functional properties.

### Bulk RNAseq sequencing

Bulk RNA sequencing was performed on *Mb1Cre*$^{+/WT}$ *Hells*$^{F/F}$ and littermate controls *Mb1Cre*$^{WT/WT}$ *Hells*$^{F/F}$. B220$^+$CD95$^+$GL7$^+$ GC B cells were FACS-sorted on a BD Aria III, 7 and 14 days post NP-CGG immunization, from four mice of each genotype. Due to their low frequency, GC B cells were FACS-sorted twice using the same gating strategy. Collected samples were lysed and processed for total RNA using the RNeasy Kit (QIAGEN), including an on-column DNAse treatment step. RNA integrity and concentration were assessed by capillary electrophoresis using Fragment Analyzer (Agilent Technologies). RNAseq libraries were prepared starting from 139.8 or 29.3 ng of total RNA (3 groups) using the Universal Plus mRNA-Seq kit (Nugen) as recommended by the manufacturer. Briefly, mRNA were captured with polyA$^+$ magnetic beads from the total RNA, chemically fragmented, and single strand and second strand cDNA were produced and then ligated to Illumina compatible adaptators with Unique Dual Index. The cDNA produced were PCR amplified after an initial test to evaluate the suitable number of PCR cycle to apply to each sample. To produce oriented or 'stranded' RNAseq libraries, a final step of strand selection was performed. An equimolar pool of the final indexed RNA-Seq libraries was prepared (the NuQuant system from Nugen was used to facilitate the RNAseq libraries quantification and normalization) and sequenced on a

NovaSeq6000 from Illumina (Paired-End reads 100 bases + 100 bases). A total of ~50 millions of passing filter paired-end reads was produced per library.

FASTQ files were mapped to the ENSEMBL *Mus musculus* GRCm38/mm10 reference using HISAT2 (v2.1.0) and counted by featureCounts from the Subread R package. Read count normalization and groups comparisons were performed by DESeq2 (v1.24.0). Flags were computed from counts normalized to the mean coverage. All normalized counts <20 were considered as background (flag 0) and >=20 as signal (flag=1). Enrichment analyses were performed using GSEA software v4.2.3 (Broad Institute) with default settings on whole transcriptome using a scoring method based on sign of fold change and Benjamini-Hochberg (BH) corrected *p* value. Heatmaps were performed on normalized read counts using pheatmap package in R. Signatures from literature were performed using analysis on GREIN (ilincs.org)[117] and selecting top 200 genes based on *p* value for genes with a log2foldchange > 1.5. R (v3.6.2) from the R project for Statistical Computing [http://www.r-project.org/]

### Single-cell RNA sequencing

Single-cell RNA sequencing of GC B cells from *Mb1Cre*$^{+/WT}$*Hells*$^{F/F}$ and littermate controls *Mb1Cre*$^{WT/WT}$*Hells*$^{F/F}$ was carried out at day 7 and 14 post NP-CGG i.p. immunization. At each time point, two mice of each genotype were used. B220$^+$CD95$^{hi}$GL7$^{hi}$ viable splenic GC B cells were single-cell FACS-sorted (AriaIII) into 384-well capture plates (Single Cell Discoveries) containing a small 50 nL droplet of barcoded primers and 10 μL of mineral oil (Sigma M8410) and immediately stored at −80 °C. Single-cell RNA sequencing was performed by Single Cell Discoveries according to an adapted version of the SORT-seq protocol[118] with primers described in ref. 119. Following amplification, library preparation was done following the CEL-Seq2 protocol[120]. The DNA library was paired-end sequenced on an Illumina Nextseq™ 500, high output, with a 1 × 75 bp Illumina kit.

Reads were mapped on *Mus Musculus* mm10 reference transcriptome, including mitochondrial genes with BWA-MEM[121]. Data was demultiplexed as described in[122]. Mapping and generation of count tables were automated using MapAndGo script (https://github.com/anna-alemany/transcriptomics/tree/master/mapandgo). Analysis of data was performed on R version 4.1.2. In total, 2304 (576 of each genotype and days post immunization) GC B cells were sequenced. Initial quality control and cell filtering was performed on Transcript Counts using the R Seurat Package version 4.1.1[123]. After removing wells where more than 50% of transcripts mapped to ERCC spike-ins, cells with <10% mitochondrial stressed associated genes, cells with <400 UMIs or 250 genes detected, 429 day 14 *Mb1Hells*$^{KO}$, 466 day 7 *Mb1Hells*$^{KO}$, 414 day 14 control and 448 day 7 control GC B cells were left for further analysis. Principal component analyses (25 pca) were performed on normalized genes and embedded in two-dimensional UMAP plots. Clustering was performed with *FindNeighbors* and *FindClusters* methods using 25 pc, and 0.5 resolution. *FindAllMarkers* methods of the Seurat Package was used to find marker genes between clusters and calculated with the Wilcoxon Rank Sum test on normalized counts. DotPlots were generated to represent average expression of selected genes of cells in different clusters on normalized counts.

### Enzymatic methyl-sequencing

Sample Collection was performed as in RNAseq bulk. Genomic DNA from sorted B lymphocytes was extracted using the Kit Quick-DNA Microprep Plus (Zymo). The 12 methyl-seq libraries (6 *Mb1*$^{Cre/WT}$ *Hells*$^{KO}$ and 6 *Mb1*$^{WT/WT}$*Hells*$^{F/F}$ littermate controls, spanning 3 time points, each one consisting of two biological replicates) were prepared using the NEBnext Enzymatic Methyl-seq kit (New England BioLabs, E7120) and sequenced as a pool of six indexed libraries in two runs on Illumina NextSeq sequencer in a 150-bases single-end read mode.

The quality of raw sequence reads was analyzed using FastQC v0.11.8 (http://www.bioinformatics.babraham.ac.uk/projects/fastqc/). Raw reads were trimmed to remove both poor-quality calls and adapters using Trim Galore (v0.6.4, www.bioinformatics.babraham.ac.uk/projects/trim_galore/, parameters: -q 20 --stringency 1 -e 0.1 --length 20). Trimmed reads were aligned to GRCm38 mouse genome assembly with Bismark v0.22.3[124] (https://www.bioinformatics.babraham.ac.uk/projects/bismark/) with default parameters. The mapping output was deduplicated with function deduplicate_bismark and default parameters, then CpG methylation calls were extracted using bismark_methylation_extractor, asking to ignore the first and last 9 bases of each read. This decision was based on the generated M-bias report files. The methylation conversion rates for all samples were obtained by mapping all reads to the spiked-in CpG unmethylated lambda and CpG methylated pUC19 DNA using the Bismark pipeline. The Pearson correlation coefficient between biological duplicates is >0.95 (Supplementary Fig. 5b).

Due to the sparsity of reads covering single cytosines, DNA methylation level was calculated by number of methylated CpG vs total covered CpG over genomic tiles spanning 50 adjacent CpG sites. The tiles were generated using SeqMonk (https://www.bioinformatics.babraham.ac.uk/projects/seqmonk/) Read Position Probe Generator, asking for a minimum of 1 read count per cytosine over all samples. SeqMonk was also used to find outlier coverage regions of the genome, by quantifying methylation over 25 kb windows and picking those regions with methylation level 10× above median. 50 CpG tiles overlapping with outlier windows were removed and the same filter was applied to promoters, non-promoter CpG Islands and repetitive elements (see below for sequence types definition).

Methylation quantitation over tiles was performed in a custom R script using function 'regionCounts' from bioconductor package methylKit v1.10.0[124], asking for a minimum of 10 bases covered per tile (cov.bases=10). With the same R function, methylation was also quantified over promoters (with cov.bases=10) and non-promoter CpG Islands (CGIs) (with cov.bases=10).

Promoter coordinates were retrieved in a custom script from Gencode vM25 annotation[125] as the 700 bp upstream and the 200 bp downstream of the start positions of all transcripts, as in ref. 126. The same Gencode annotation was also used to annotate tiles as exons, introns or intergenic regions for the violin plots of Supplementary Fig. 5c, using bioconductor package genomation v1.16.0[127]. Tiles annotated as intergenic were additionally filtered to remove all the ones also overlapping a repetitive element, even just with 1 bp. To obtain non-promoter CGIs, UCSC Table Browser track form mm10 was downloaded as a bed file and those islands with >50% overlap with the 900 bp promoter regions were removed. Promoters were divided in a High CpG and Low CpG subset in a custom script, as follows: the normalized CpG fraction (observed number of CpG divided by expected number of CpG) was computed for all promoters as in ref. 126; the distribution of these values was fitted by two Gaussian distributions and the intersection between the two was used as the normalized CpG fraction threshold to separate the two types of promoters.

To quantify CpG methylation over repetitive elements, Repeat Library 20140131 (mm10, Dec 2011) was downloaded from the RepeatMasker website (A.F.A. Smit, R. Hubley & P. Green RepeatMasker at http://repeatmasker.org) in fasta format, converted into bed format and then filtered by removing both all elements overlapping a gene body of Gencode vM25 and all "Simple repeats" and "Low complexity repeats". Methylation was counted over the remaining elements with 'regionCounts' (with cov.bases=10).

For correlation with expression, methylation was counted separately on full-length RNA transposable elements only, defined as >5 kb for LINE1 elements, >6 kb for IAP families and >4.5 kb for MMERVK10C, also using 'regionCounts' with cov.bases = 10.

For all methylation analyses, the methylation percentage of the two biological replicates for each tile/promoter/CGI/repetitive element was averaged. The number of reads and coverage for each replicates is provided in Supplementary Information.

## RNA-seq analysis for transposable elements (TE) expression

The Paired-end RNA-seq fastq files obtained as described above (Bulk-RNAseq) were analyzed using FastQC v0.11.8 (http://www.bioinformatics.babraham.ac.uk/projects/fastqc/). Raw reads were trimmed to remove both poor-quality calls and adapters using Trim Galore (v0.6.4, www.bioinformatics.babraham.ac.uk/projects/trim_galore/, parameters: -q 20 --stringency 1 -e 0.1 --length 20 –paired). Trimmed reads were aligned to GRCm38 mouse genome with STAR v2.7.5c, with parameters recommended by[128] for the analysis of transcripts derived from autonomous transposable elements (TEs) in the mouse genome. After alignment, using a custom script, only reads from pairs where both mates were completely included into a repetitive element and not overlapping with gene bodies of Gencode vM25 were kept. As for methylation analysis, Repeat Library 20140131 (mm10, Dec 2011) was used as repetitive elements annotation, after excluding "Simple repeats" and "Low complexity repeats".

With the STAR parameters above, only one random alignment (the best one found) is reported for multimappers, preventing the precise quantification of single repetitive elements. Therefore, for each repName of the repetitive element annotation—often present in multiple copies in the genome—reads counts were summarized using featureCounts[129], a function of the Subread program (v2.0.1, http://subread.sourceforge.net), with parameters -p -B -s 0 --fracOverlap 1 -M. The downstream analysis of TEs expression was performed in a custom Rmd script. First, an analysis at family level (repFamily field in the annotation) was performed in order to identify samples with an important contamination from genomic DNA. The latter can affect the quantification of transposable elements transcripts. For this analysis, FPKM were computed for each repFamily and FPKM values of DNA transposon families were manually checked. The three samples (naives_344, naives_440 and J14_394) showing high DNA contamination were excluded from the differential expression (DE) analysis of RNA TEs. DE analysis of RNA TEs was performed between control and Mb1Hells[KO] samples separately for each time point, using bioconductor package DESeq2 v1.24.0[130]. The test used for statistical significance was the Wald test and the significance cutoff for optimizing the independent filtering was 0.05. The MA plots in Fig. 5c show the shrunken log2 fold change, obtained using 'ashr' method[131].

For correlation with methylation, a separate featureCounts command and DESeq function was run only with full-length RNA TEs, defined as >5 kb for LINE1 elements, >6 kb for IAP families and >4.5 kb for MMERVK10C. For Fig. 5c, only full-length elements with baseMean expression values > 20 are shown, since very lowly expressed ones would erroneously look deregulated due to noise affecting their low read counts.

We used Integrative Genomics Viewer software from Broad Institute (IGV, version 2.8.2) to visualize derepressed IAPEz-int copies in the vicinity of selected genes, and splice junctions (Sashimi plots).

## Statistics and reproducibility

No statistical method was used to predetermine sample size. The experiments were not randomized. The investigators were not blinded to allocation during experiments and outcome assessment. No data were excluded from the analyses. For comparison of means, statistical analyses were performed using GraphPad Prism 9.4.1 software. Unpaired two-tailed t-test were performed in this study, with Welch's correction in case of unequal variances.

**Reporting summary**

Further information on research design is available in the Nature Portfolio Reporting Summary linked to this article.

## Data availability

The bulk RNAseq data generated in this study have been deposited in the Array Express database under accession code E-MTAB-12638. The single cell RNAseq data generated in this study have been deposited in the Array Express database under accession code E-MTAB-12499. The EM-seq data generated in this study have been deposited in the Array Express database under accession code E-MTAB-12609. Other RNAseq data used in this study are available in the GEO database under accession codes GSE109125, GSE158605, GSE141423, GSE60927, GSE115656, GSE89897, GSE110669 and GSE128710. *Mus musculus* genome assembly mm10 is available through the hyperlink https://www.ncbi.nlm.nih.gov/datasets/genome/GCF_000001635.20/. All other original data from this study are included in the text. Source data are provided with this paper.

## Code availability

All original code has been deposited at GitHub and is publicly available under the DOIs: https://zenodo.org/record/8208681[132] and https://zenodo.org/record/8208644[133] and the following links: https://github.com/boulardlab/BS_RNASeq and https://github.com/boulardlab/BS_EMSeq.

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

## Acknowledgements

This research was supported by a grant from the Agence Nationale de la Recherche (Methyl-Memory, ANR-14-CE14-0011 (C.-A.R.)), by the Fondation Princesse Grace and by EMBL. CC and EM were supported by doctoral fellowships from the french Ministère de l'Enseignement Supérieur, de la Recherche et de l'Innovation, at *Ecole Doctorale* BioSPC ED562, Université Paris Cité. This project has been funded in part with federal funds from the National Cancer Institute, National Institutes of Health, under contract HHSN26120080001E (K.M.). The content of this publication does not necessarily reflect the views or policies of the Department of Health and Human Services, nor does mention of trade names, commercial products, or organizations imply endorsement by the U.S. Government. This Research was supported by the Intramural Research Program of the NIH, National Cancer Institute, Center for Cancer Research. We thank the cytometry, the genomic, and the LEAT platforms of the Structure Fédérative de Recherche SFR-Necker, and the EMBL Genomic Core facility for EM-seq. We thank Nicolas Cagnard and Pascal Chappert for their help for bulk and scRNAseq analyses respectively, and Paola Sanna and Maxime Puès for technical assistance. We thank Jonatan Ersching for advice on pS6 staining, and Sandra Weller for critical reading of the manuscript. We thank GSK and Dr. Melissa Pappalardi for providing the GSK3685032A DNMTi.

## Author contributions

C.C. and E.M. designed, performed, analyzed, and interpreted experiments on mice and cells, assisted by A.D.S., D.L. and S.S. C.C. also analyzed scRNA-seq data, performed GSEA on bulk RNAseq and drew figures. S.F. analyzed EM-seq experiments and methylome/transcriptome correlations, supervised by M.B. A.D.S. and S.F. contributed equally. K.M. and J.R. provided *Hells*-floxed mice. J-C.W., C.-A.R., and S.S. analyzed and interpreted the data. S.S. conceived the study and wrote the manuscript, with critical feedback from C.C., S.F., M.B., J.-C.W. and C.-A.R.

## Competing interests

J-C.W. received consulting fees from the Fondation Mérieux outside of this work. The remaining authors declare no competing interests.
