## [Peer Review File · Nature Communications]

Germinal center output is sustained by HELLS-dependent DNA-methylation-maintenance in B cellsREVIEWER COMMENTS

Reviewer #1 (Remarks to the Author):

In this manuscript, the authors generated B-cell-specific *Hells* conditional knockout mice and showed that *Hells* deficiency induces an accelerated decay of germinal center (GC) B cells and impairs the generation of high affinity memory B cells and antibodies. The authors also showed that mutant GC B cells undergo drastic DNA hypomethylation and derepression of retrotransposons. Therefore, *Hells*-mediated maintenance DNA methylation is likely essential for GC B cell persistence. On the other hand, it is known that about half ICF syndrome patients have mutations in de novo DNA methyltransferase *DNMT3B* (ICF1), and another half have mutations either in *ZBTB24* (ICF2), *CDCA7* (ICF3), or *HELLS* (ICF4), which are involved in maintenance DNA methylation and NHEJ in DNA repair and immunoglobulin class switch recombination. All ICF patients show immunodeficiency and DNA hypomethylation, but it has been a mystery why mutations in *DNMT3B*, which seems not to be involved in NHEJ, cause immunodeficiency. I think that the authors' finding could explain the mystery. Although the detailed mechanism how DNA methylation contribute to the GC B cell persistence remains unclear, findings made by the authors are very important and thus the manuscript is suitable for publishing from Nature Communications with some revisions.

Major points

1. The authors deleted 327 bp [exon 9 (183 bp) and exon 10 (144 bp)] of the *Hells* gene, which encodes 109 aa. Since mouse *Hells* is composed of 821 aa, the "KO" mice could express short form of *Hells* composed of 712 aa (It is highly possible that mRNA from the "KO" allele could escape from non-sense mediated decay since the deleted exons do not include the last exon). Therefore, the authors should examine whether their *Hells* KO mice express short form of hypomorphic *Hells* protein by Western blotting. If the short form expresses, to avoid misunderstanding ("KO" usually indicates "null"), the authors should change the word "*Hells*-KO mice" to "*Hells* mutant (or Δ SNF2) mice" and revise all related descriptions and discussions (*Hells*-KO B cells might not die because the cells express hypomorphic *Hells*).
2. Fig.5: Compared with peripheral blood leukocytes of ICF4 patients (Velasco et al., Int J Mol Sci, 2021; Unoki et al., Hum Mol Genet, 2022), DNA hypomethylation occurred in *Hells*-KO GC B cells is drastic. It has been shown that the *CDCA7*/*HELLS* complex is required for chromatin remodeling of heterochromatic and late replicating regions where DNA methylation is maintained in a replication uncoupled manner (Velasco et al. Hum Mol Genet, 2019; Ming et al., Cell Research). The authors' finding suggests several possibilities such as (1) chromatin remodeling by the *CDCA7*/*HELLS* complex is somehow required for maintenance DNA methylation of wider genomic regions in *Hells*-KO GC B cells, (2) *Hells* knockout activates TET-mediated DNA demethylation pathway in the cells, and (3) *Hells* helps maintenance DNA methylation by *Dnmt3B* in the cells (see line 377). The possibility of activation of TET-mediated DNA demethylation pathway may be examined by immunofluorescence using anti-5hmC antibody. Since the authors performed RNA-seq, expression levels of TETs, DNMTs, UHRF1, and ICF related genes in *Hells*-KO GC B cells are also easily checked and the information may help evaluating these possibilities.
3. Fig. 8: The authors subcutaneously injected GSK3685032 DNMT1 inhibitor. Therefore, DNA hypomethylation could occur all cells in entire body. I concern secondary effects. Firstly, the authors should check whether DNA hypomethylation was induced in GC B cells by the treatment (classical methods are fine). Secondly, the authors should check whether expression levels of *HELLS*, *CDCA7*, and *ZBTB24* were unchanged in GC B cells by the treatment. If any, the authors should describe adverse effects such as loss of body weight.
4. Line 373: The authors wrote "This may seem paradoxical because the de novo DNA methyltransferase *DNMT3B* is most frequently mutated in ICF patient. However, a complete impairment of de novo methylation activity in the murine B-lineage through genetic deletion of both *Dnmt3a* and *Dnmt3b* leads to an opposite phenotype ... (Barwick et al., 2018)." Barwick et al used *Cd19-Cre* to knockout de novo *Dnmts*. The authors should realize that establishment of DNA methylation pattern during post-implantation development by de novo *Dnmts* is impaired in ICF1 patients. After establishment of DNA methylation pattern, KO effect of de novo *Dnmts* can be minimum since DNA methylation pattern is maintained by the *DNMT1*/*UHRF1* complex. Because of this and since some hypomethylated regions are common to all ICF patients (Velasco et al., Hum Mol Genet, 2018), I do not think that it is paradoxical. As I described above, it has been a mystery

why ICF1 patients show immunodeficiency. I think that the authors' finding, of which maintenance DNA methylation is essential for GC B cell persistence, can explain the mystery and thus would like the authors to discuss about it deeply.

5. Line 398: The authors wrote "It is noteworthy that genome-wide hypomethylation affected neither GC B cell proliferation nor survival, in stark contrast with the cell-cycle arrest caused by Uhrf1 loss-of function in the same cells (Chen et al., 2018). The authors should realize that Chen et al used AID-Cre to knockout UHRF1, which results in complete loss of DNA methylation. Although Hells KO induced dramatic loss of DNA methylation, DNA methylation is not completely lost in the cells (It is clear in Extended Data Figure 5A). Probably, the phenotypical difference could be attributable to the differential methylation status between the KO cells.

Minor points

1. Some descriptions regarding HELLS in "Introduction" are insufficient.

1) The authors should include previous findings that the CDCA7/HELLS complex is involved in NHEJ for DNA repair and immunoglobulin class switch recombination (Unoki et al., J Clin Invest, 2019; He et al., PNAS, 2020).

2) The authors wrote "Recently, a model was proposed to integrate all four proteins in a common DNA methylation pathway... (Jenness et al., 2018). However, currently, the model is not widely accepted. Current consensus is that the CDCA7/HELLS complex recruits the DNMT1/UHRF1 maintenance DNA methylation complex, not DNMT3B, to newly synthesized DNA strands to maintain DNA methylation pattern (Ming et al., Cell Research, 2020; Han et al., Nucleic Acids Res, 2020; Unoki et al., Sci Rep, 2020).

3) The authors wrote "However, the genome-wide distribution of differentially methylated sequences is quite different between ICF1, and ICF2, 3 and 4 patients...." Molecular mechanism behind generation of the difference has already been well examined by many groups (Velasco et al. Hum Mol Genet, 2019; Unoki et al. Sci Rep, 2020; Han et al. Nucleic Acids Res, 2020; Ming et al. Cell Research, 2020; Nishiyama et al. Nat Commun, 2020), and their findings are well summarized in a review paper (Unoki et al., Genes Cells, 2021). The authors should include these important findings in their introduction.

2. Line 144-150: "Fig. 3d" appears before "Fig. 3c" in the main text. Please check the order of figure numbers through the manuscript.

3. Line 396: The authors wrote "..., which suggests that cytosolic sensing of nucleic acids is physiologically mitigated in these cells." I would like to know expression levels of factors involved in the cytosolic sensing of nucleic acids (RNA-seq data are fine).

4. The authors need to add explanations of abbreviations such as MFI, FDR, and NES in figure legends.

Reviewer #2 (Remarks to the Author):

This manuscript from Cousu et al explores the role and mechanism of HELLS in maintaining normal DNA methylation in germinal centre B cells. Loss of function for HELLS causes the ICF4 immunodeficiency, and this manuscript provides a mechanism that underlies the defects in humoral immunity for these patients. With elegant mouse knockout models, thorough flow cytometry-based immunophenotyping and high quality genomic analyses, this is an excellent and fascinating study that provides significant and novel insights into the epigenetic regulation of the GC reaction – congratulations to the authors. The profound loss of DNA methylation in HELLS knockout mouse is incredibly striking and the proposed model whereby this predisposes GC B cells to exit and differentiate is compelling. In my opinion, this study will be of interest to B cell immunologists and genomic researchers alike as it offers some truly unique observations.

I have only minor comments and suggestions below.

- Can the authors expand upon the potential link between HELLS expression and cellular proliferation in the GC? Looking at the expression of HELLS in human GC B cells at <https://www.tonsilimmune.org/> showed that the enrichment of HELLS expression is most striking in cycling/proliferating GC B cells. If HELLS is differentially regulated through the cell cycle, this has some implications for its mechanism in ICF4.

- Figure 3a – some inconsistent visualization. Show all data points like in 3C

- Interesting that IgA switching does not seem to be affected. Are there insights about this from the methylation analyses at the IgH locus? Or potential transcription factors that could mediate this, compared to the defects in IgG
- Line 171-172 – the reference to “GC physiology” should be clarified, as these conclusions are only assessed by FACS (ie cells in suspension) of light/dark zone B cells and no histology is included. The addition of “as assessed by flow cytometry” would be sufficient.
- How do the classes of IAP and other repetitive elements that are activated upon HELLS/DNA methylation loss compare with other experimental systems that see similar results upon DNMT knockout. Are they similar to those that reactivated in embryonic stem cells and neurons for example, or are some subsets of repetitive elements reactivated specific to B cells? One potential resource to consider is Kaluscha, S., et al Nat Genet 54, 1895–1906 (2022) <https://doi.org/10.1038/s41588-022-01241-6>
- line 233-234 - please reword as the relevance of “non-immunological tissue-specific TFs” is a little unclear. Potentially add the descriptor of “non-immunological tissue-specific TFs whose sequence motifs are in the IAP repetitive elements”
- line 237 – reword “illegitimate” – too subjective
- It is not clear how the preMBC transcriptional signature was defined.
- Did the authors examine pre-MBC markers in the scRNAseq dataset? Looking at either individual markers or the pre-defined preMBC transcriptional signature would be valuable. Similarly, if possible, can the authors detect reactivated repetitive elements in this dataset? This would be a valuable (but not essential) addition to the analysis to see if only the HELLSKO-specific B cell cluster/state has the reactivation, or if all B cell subsets do.
- P323 – please define ISR

Reviewer #3 (Remarks to the Author):

This well performed study examines the impact of HELLS (Helicase, lymphoid specific) deficiency in the B cell lineage on the antibody response. HELLS is shown to be highly expressed by germinal center (GC) B cells in mice and humans. B lineage KO of HELLS using two different Cre drivers is shown to cause a gradual loss of GC B cells, a lower IgG response and less efficient affinity maturation. Memory B cell generation and plasma cell generation are both reduced. GC B cell viability appeared not to be affected but instead there was evidence for accelerated differentiation towards memory and plasma cells. GC B cells from HELLS cKO mice had markedly reduced DNA methylation and this was associated with de-repression of non-coding repeats, transposable elements, and a small subset of genes. scRNAseq identified a cluster of KO GC B cells that had elevated ATF4 dependent integrated stress response (ISR) pathway activity, including elevated CD98 amino acid transporter expression. This activity is suggested to contribute to premature acquisition of plasma cell features in GC cells. Finally, treatment of mice with the DNMT1 inhibitor GSK 3685032A led to similar effects on GC B cells as observed in the cKO mice.

Overall, this study provides valuable insight regarding the function of HELLS in the B cell response and helps understand how loss-of-function mutations can contribute to humoral immunodeficiency in ICF4 syndrome. The work is likely to be of broad interest. I have only a few minor concerns.

1. It should be indicated whether the control mice for the Mb1Cre HELLS KOs were Mb1Cre+ or not since these mice are Mb1 heterozygote and this can have effects on BCR signaling. While this information is important to provide, this is not a major concern as similar findings were made using a second Cre drive (CD21Cre). Some differences were noted in the antibody response between the two crosses – some comment should be made about these differences.
2. The basis for the marked loss of GC B cells over time remains unclear given that there is no observed increase in apoptosis, and strongly reduced generation of mature memory B cells and plasma cells. The notable decrease in FADS2 expression makes one wonder about increased ferroptosis. It would be helpful if a positive control for cells undergoing ferroptosis could be provided to ensure the assay is working.
3. Another explanation for the reduction in GC B cells could be a reduced proliferation rate. Given the very rapid proliferation of GC B cells, small differences can lead to big effects. The data in

Figure 4c show a trend toward more cKO cells in G1 and less in S. This analysis was done at day 10, when the effect on GC size seems variable. This analysis should be shown for a second, later time point.

4. In figure 1 it is stated that Hells is overexpressed in GC B cells. This wording should be changed to indicate that Hells is more highly expressed in GC compared to naïve or follicular B cells. (Overexpression implies something non-physiological).

5. Line 293, design should be designate.

Major points

1. The authors deleted 327 bp [exon 9 (183 bp) and exon 10 (144 bp)] of the *Hells* gene, which encodes 109 aa. Since mouse *Hells* is composed of 821 aa, the “KO” mice could express short form of *Hells* composed of 712 aa (It is highly possible that mRNA from the “KO” allele could escape from non-sense mediated decay since the deleted exons do not include the last exon). Therefore, the authors should examine whether their *Hells* KO mice express short form of hypomorphic *Hells* protein by Western blotting. If the short form expresses, to avoid misunderstanding (“KO” usually indicates “null”), the authors should change the word “*Hells*-KO mice” to “*Hells* mutant (or Δ SNF2) mice” and revise all related descriptions and discussions (*Hells*-KO B cells might not die because the cells express hypomorphic *Hells*).

As anticipated by reviewer #1, activated B cells from *Mb1-Cre Hells^{F/F}* mice indeed express a stable mRNA with an in-frame deletion of exons 9 and 10. However, we could not detect the expected 82 kDa truncated protein by western-blot, despite the use of a polyclonal antibody that targets the C-terminal domain encoded by exons downstream the deletion. We therefore think that the use of the term “KO” is acceptable in this context.

The electrophoresis which shows the RT-PCR products, the electrophoregram of the truncated mutant RNA and the western-blot for HELLS are shown in the revised Supplementary Fig. 2d,e,f, and described in the results part (lines 118-122). The corresponding methods have also been added (lines 545-555 and 642-646).

2. Fig.5: Compared with peripheral blood leukocytes of ICF4 patients (Velasco et al., *Int J Mol Sci*, 2021; Unoki et al., *Hum Mol Genet*, 2022), DNA hypomethylation occurred in *Hells*-KO GC B cells is drastic. It has been shown that the CDCA7/HELLS complex is required for chromatin remodeling of heterochromatic and late replicating regions where DNA methylation is maintained in a replication uncoupled manner (Velasco et al. *Hum Mol Genet*, 2019; Ming et al., *Cell Research*). The authors’ finding suggests several possibilities such as (1) chromatin remodeling by the CDCA7/HELLS complex is somehow required for maintenance DNA methylation of wider genomic regions in *Hells*-KO GC B cells, (2) *Hells* knockout activates TET-mediated DNA demethylation pathway in the cells, and (3) *Hells* helps maintenance DNA methylation by *Dnmt3B* in the cells (see line 377). The possibility of activation of TET-mediated DNA demethylation pathway may be examined by immunofluorescence using anti-5hmC antibody. Since the authors performed RNA-seq, expression levels of TETs, DNMTs, UHRF1, and ICF related genes in *Hells*-KO GC B cells are also easily checked and the information may help evaluating these possibilities.

We agree that all three hypotheses described by reviewer #1 may explain the severe hypomethylation observed in the mutant mice. We did not observe in our RNAseq data any significant change in expression of *Tet1*, *Tet2*, *Tet3*, *Dnmt1*, *Dnmt3a*, *Dnmt3b*, *Uhrf1*, *Zbtb24* and *Cdca7* in *Mb1Hells^{KO}* GC B cells, as depicted for day 14 in the figure below :

We then assessed 5hmC level by two methods :

1. Flow cytometry on day 12 GC B cells, with a rabbit anti-5hmC serum, following the protocol described in Chen et al., Genes Dev. 2013 27(18):1974-85.

Mb1Hells^{KO} GC B cells did not show significant increase in fluorescence, as shown in the figure below :

Left : flow cytometry on day 12 GC B cells from a control mouse, stained with anti-5hmC and an AF647-conjugated anti-rabbit IgG (solid line), or with the secondary antibody alone (grey area).

Right : quantification by flow cytometry of 5hmC amount on control or *Mb1Hells*^{KO} day 12 GC B cells (n=7 for each genotype).

2. ELISA on day 12 GC B cells with Global 5-hmC DNA ELISA Kit from Active motif (ref 55025).
5hmC level was below the detection limit for all samples

The mechanism of DNA methylation loss in the absence of HELLS in GC B cells is now more extensively discussed in the revised version of the discussion (lines 414-425).

3. Fig. 8: The authors subcutaneously injected GSK3685032 DNMT1 inhibitor. Therefore, DNA hypomethylation could occur all cells in entire body. I concern secondary effects. Firstly, the authors should check whether DNA hypomethylation was induced in GC B cells by the treatment (classical methods are fine). Secondly, the authors should check whether expression levels of HELLS, CDCA7, and ZBTB24 were unchanged in GC B cells by the treatment. If any, the authors should describe adverse effects such as loss of body weight.

The toxicity of the GSK3685032 compound has been studied extensively in the seminal paper from Pappalardi (Pappalardi, 2021 ; réf 86) in which mice showed no weight loss after 28 days of treatment with the same dose that we used (30 mg.kg⁻¹), nor any alteration of behavior and grooming. In our own study, mice treated with GSK3685032 did not lose weight after 5 days of treatment (see figure below).

This absence of toxicity is now emphasized in the results part (lines 377-379), but we chose not to include the data on the absence of weight loss after treatment as this has been published earlier (ref 86).

We checked the effect of GSK3685032 DNMT1 inhibitor on DNA hypomethylation in day 14 GC B cells (4 mice treated with the inhibitor and 4 mice treated with the vehicle), by analyzing CpG methylation of LINE-1 repeats through targeted bisulfite-sequencing. As expected, treatment with GSK3685032 increased significantly the proportion of unmethylated CpG sites in LINE-1 repeats (18.2% of 713 CpG sites with GSK3685032, vs. 7.9% of 719 CpG sites with the vehicle ; $p=0.0014$).

The level of hypomethylation is lower than that observed in day 14 *Mb1Hells*^{KO} GC B cells, but :

- (1) the treatment was only administered from day 7 to day 13, while *Hells* deletion starts during early B-cell ontogeny in *Mb1Hells*^{KO} mice and persists throughout B-cell activation
- (2) the magnitude of hypomethylation is comparable to that observed in tumor cells from animals treated with GSK3685032 in the seminal article from Pappalardi and colleagues (ref 86)

These data have been added in Fig. 8c and are now described in the results part (lines 380-382).

We checked the expression of *Hells*, *Cdca7* and *Zbtb24* by qRT-PCR on day 12 B220⁺GL7^{hi}CD95^{hi} cells from mice treated for 5 days with GSK3685032 DNMT1 inhibitor or with the vehicle. Based on the comparison with two different housekeeping genes, we conclude that none of the three genes displays important up- or down-regulation (figure below).

We thus favor the hypothesis that the hypomethylation consecutive to treatment with GSK3685032 is likely a direct consequence of DNMT1 inhibition, and not a side-effect caused by downregulation of other genes in DNA methylation.

We did not include these data in the article as the *in vivo* toxicity of GSK3685032 is limited, which points to a causal link between DNMT1 inhibition by the compound and DNA hypomethylation.

4. Line 373: The authors wrote “This may seem paradoxical because the *de novo* DNA methyltransferase DNMT3B is most frequently mutated in ICF patient. However, a complete impairment of *de novo* methylation activity in the murine B-lineage through genetic deletion of both *Dnmt3a* and *Dnmt3b* leads to an opposite phenotype ... (Barwick et al., 2018).” Barwick et al used *Cd19-Cre* to knockout *de novo* *Dnmts*. The authors should realize that establishment of DNA methylation pattern during post-implantation development by *de novo* *Dnmts* is impaired in ICF1 patients. After establishment of DNA methylation pattern, KO effect of *de novo* *Dnmts* can be minimum since DNA methylation pattern is maintained by the DNMT1/UHRF1 complex. Because of this and since some hypomethylated regions are common to all ICF patients (Velasco et al., Hum Mol Genet, 2018), I do not think that it is paradoxical. As I described above, it has been a mystery why

ICF1 patients show immunodeficiency. I think that the authors' finding, of which maintenance DNA methylation is essential for GC B cell persistence, can explain the mystery and thus would like the authors to discuss about it deeply.

We thank reviewer #1 for this highly relevant comment which helped us improve our discussion on the role of *de novo* DNA methylation and DNA methylation maintenance in ICF syndrome.

We have modified our discussion accordingly (lines 410-438).

5. *Line 398: The authors wrote "It is noteworthy that genome-wide hypomethylation affected neither GC B cell proliferation nor survival, in stark contrast with the cell-cycle arrest caused by Uhrf1 loss-of function in the same cells (Chen et al., 2018). The authors should realize that Chen et al used AID-Cre to knockout UHRF1, which results in complete loss of DNA methylation. Although Hells KO induced dramatic loss of DNA methylation, DNA methylation is not completely lost in the cells (It is clear in Extended Data Figure 5A). Probably, the phenotypical difference could be attributable to the differential methylation status between the KO cells.*

Reviewer #1 is right to point to potential differences in the magnitude of hypomethylation in the two mouse models, **which has been included in the revised discussion (line 451-452).**

Minor points

1. *Some descriptions regarding HELLS in "Introduction" are insufficient.*

1.1 *The authors should include previous findings that the CDCA7/HELLS complex is involved in NHEJ for DNA repair and immunoglobulin class switch recombination (Unoki et al., J Clin Invest, 2019; He et al., PNAS, 2020).*

The role of CDCA7/HELLS complex in NHEJ and the corresponding references have been added to the introduction (lines 78-81)

1.2. *The authors wrote "Recently, a model was proposed to integrate all four proteins in a common DNA methylation pathway.... (Jenness et al., 2018). However, currently, the model is not widely accepted. Current consensus is that the CDCA7/HELLS complex recruits the DNMT1/UHRF1 maintenance DNA methylation complex, not DNMT3B, to newly synthesized DNA strands to maintain DNA methylation pattern (Ming et al., Cell Research, 2020; Han et al., Nucleic Acids Res, 2020; Unoki et al., Sci Rep, 2020).*

1.3 *The authors wrote "However, the genome-wide distribution of differentially methylated sequences is quite different between ICF1, and ICF2, 3 and 4 patients...." Molecular mechanism behind generation of the difference has already been well examined by many groups (Velasco et al. Hum Mol Genet, 2019; Unoki et al. Sci Rep, 2020; Han et al. Nucleic Acids Res, 2020; Ming et al. Cell Research, 2020; Nishiyama et al. Nat Commun, 2020), and their findings are well summarized in a review paper (Unoki et al., Genes Cells, 2021). The authors should include these important findings in their introduction.*

The revised version of the introduction includes these updated data and the corresponding references about the role of the different proteins involved in ICF syndrome, as well as the consequences of their loss-of-function on DNA methylation in ICF patients (lines 70-78).

2. Line 144-150: "Fig. 3d" appears before "Fig. 3c" in the main text. Please check the order of figure numbers through the manuscript.

The order of appearance of Fig. 3c, d and e has been corrected in the corresponding figure and figure legends.

3. Line 396: The authors wrote "..., which suggests that cytosolic sensing of nucleic acids is physiologically mitigated in these cells." I would like to know expression levels of factors involved in the cytosolic sensing of nucleic acids (RNA-seq data are fine).

We apologize for the omission of the reference to Supplementary Fig. 8b which depicted the expression levels of factors involved in nucleic acids sensing during late B-cell differentiation (now Fig. 8c). We added an extended analysis of the expression of these molecules in our own RNAseq data (Supplementary Fig. 8b), which confirms the global repression of these sensors in day 7 and day 14 GC B cells, relative to naive follicular B cells.

4. The authors need to add explanations of abbreviations such as MFI, FDR, and NES in figure legends.

Abbreviations have been added to all figures.

Point-by-point response to Reviewer #2 (Remarks to the Author):

1. Can the authors expand upon the potential link between HELLS expression and cellular proliferation in the GC? Looking at the expression of HELLS in human GC B cells at <https://www.tonsilimmune.org/> showed that the enrichment of HELLS expression is most striking in cycling/proliferating GC B cells. If HELLS is differentially regulated through the cell cycle, this has some implications for its mechanism in ICF4.

As noted by reviewer #2, human *Hells* is indeed expressed predominantly in the cycling population of GC B cells, according to the article of King and colleagues (King et al., 2021). We looked at *Hells* expression in our scRNAseq data, after separation of control GC B cell populations according to cell cycle scoring from the Seurat Package in R. *Hells* mRNA is essentially present in the S-phase population (see above), which is in agreement with previous observations that indicated that "the peak of *Lsh* mRNA and protein correlated closely with the onset of S phase in ConA activated splenocytes" (Geiman and Muegge, 2000, ref 93).

Interestingly, human *Dnmt1*, *Uhrf1* and *CdcA7* show a similar pattern of expression in the cycling population of GC B cells (King et al., 2021).

We therefore added to the text of the results part the reference to King *et al.* (ref 48), to underline the expression of HELLS and CDCA7 during proliferative stages of B cell development and activation (lines 103-104). We also briefly commented on the relationship between Hells and cell cycle phase in the discussion, without showing the scRNAseq analysis presented above since it matches previously published data (ref 93, line 418-420). More generally, we now discuss extensively the role of HELLS for DNA methylation maintenance in proliferating cells in the revised versions of the introduction (lines 73-76) and of the discussion (lines 410-425).

2. *Figure 3a – some inconsistent visualization. Show all data points like in 3C*

All data points are now shown for all histograms in Fig. 3 (and all other figures).

3. *Interesting that IgA switching does not seem to be affected. Are there insights about this from the methylation analyses at the IgH locus? Or potential transcription factors that could mediate this, compared to the defects in IgG*

We thank reviewer #2 for this comment that enabled us to detect an error in the statistical analysis of the comparison of IgA levels in control and *CD21Hells^{KO}* (Supplementary Fig. 3a). We thus confirm that IgA level is not significantly altered in either of the two *Hells*-deficient mouse strains, **and corrected Supplementary Fig. 3a accordingly.**

We have no satisfactory explanation for the differential impact of *Hells* loss-of-function on IgA and IgG isotypes. Unfortunately, our RNAseq and EM-seq data were generated on splenic GC B cells induced by immunization with alum-absorbed T-dependent antigen, which biases the response toward IgG1, whereas spontaneous IgA switching and secretion likely occurs in mucosal-associated lymphoid tissues which were not included in our study. Our datasets are thus not appropriate for the study of methylation and transcription of the IgA locus in particular.

4. *Line 171-172 – the reference to “GC physiology” should be clarified, as these conclusions are only assessed by FACS (ie cells in suspension) of light/dark zone B cells and no histology is included. The addition of “as assessed by flow cytometry” would be sufficient.*

We changed the sentence accordingly (lines 189-190).

5. *How do the classes of IAP and other repetitive elements that are activated upon HELLS/DNA methylation loss compare with other experimental systems that see similar results upon DNMT knockout. Are they similar to those that reactivated in embryonic stem cells and neurons for example, or are some subsets of repetitive elements reactivated specific to B cells? One potential resource to consider is Kaluscha, S., et al Nat Genet 54, 1895–1906 (2022) <https://doi.org/10.1038/s41588-022-01241-6>*

Walsh and colleagues were the first to report that global loss of DNA methylation reactivates the transcription of the endogenous retroviruses of the family intracisternal A particle (IAP) (Nature Genetics, 1998 (20):116–117). Subsequent unbiased surveys of all families of retrotransposons transcribed in demethylated cells have showed that the IAP family is predominantly expressed (Kaluscha, Nat Genet 2022 (54):1895-1906 and references inside : Dahlet, Nat Comm 2020 19;11(1):3153; Sharif, Cell Stem Cell 2016 7;19(1):81-94; Ramesh, Genes Dev 2016, 30(19):2199-2212). In good agreement, our study shows that IAP elements are the main type of retrotransposons reactivated in demethylated GC B cells.

We emphasized the similarity between our observation and those described in Kaluscha et al. by adding a dedicated sentence in the revised results part (lines 231-232).

6. line 233-234 - please reword as the relevance of “non-immunological tissue-specific TFs” is a little unclear. Potentially add the descriptor of “non-immunological tissue-specific TFs whose sequence motifs are in the IAP repetitive elements”

The sentence has been edited for clarity (line 255).

7. line 237 – reword “illegitimate” – too subjective

We have reformulated the sentence in a more neutral way (line 258).

8. It is not clear how the preMBC transcriptional signature was defined.

The definition of the pre-MBC transcriptional signature has been clarified in the results part (lines 284-286) in addition to the methods part (lines 748-750). We added a Source data file to the submission that includes the gene list composing the pre-MBC signature.

9. Did the authors examine pre-MBC markers in the scRNAseq dataset? Looking at either individual markers or the pre-defined preMBC transcriptional signature would be valuable.

We acknowledge that this analysis would have been valuable, but the pre-MBC population was excluded from the cell sorting gate (lines 322-323) as we had already identified it as a separate population by flow cytometry; the corresponding pre-MBC signature is therefore hardly detectable in the scRNAseq dataset (not shown).

10. Similarly, if possible, can the authors detect reactivated repetitive elements in this dataset? This would be a valuable (but not essential) addition to the analysis to see if only the HELLSKO-specific B cell cluster/state has the reactivation, or if all B cell subsets do.

From our discussion with Single Cell Discoveries, it appears that transposable element analysis in scRNAseq data is fairly complicated. To answer this interesting question, we chose as a proxy to analyze the expression of genes that appeared derepressed in our bulk RNAseq data as a consequence of the presence of an IAPEz-int element in the vicinity. In our scRNAseq dataset, the expression of such genes was particularly elevated in the *Mb1Hells*^{KO} cluster, which suggests that IAPEz-int derepression is also maximal in this cluster.

This novel analysis is presented in Supplementary Fig. 7b and described in the results part (lines 328-332).

11. P323 – please define ISR

ISR definition has been added after its first appearance in the results part (line 351).

Point-by-point response to Reviewer #3 (Remarks to the Author):

1. It should be indicated whether the control mice for the *Mb1Cre* HELLS KOs were *Mb1Cre*⁺ or not since these mice are *Mb1* heterozygote and this can have effects on BCR signaling. While this information is important to provide, this is not a major concern as similar findings were made using a second *Cre* drive (*CD21Cre*). Some differences were noted in the antibody response between the two crosses – some comment should be made about these differences.

We agree with reviewer #3 that the heterozygosity of *Mb1-Cre* mice for *CD79a* might have some effect on BCR signaling; as littermate control mice used in most experiments were actually devoid of the *Cre*

allele, this is now specified in the results part (line 113-114), and the occasional usage of *Mb1^{Cre/WT}* mice as littermate controls is now indicated in the figure legends.

We think however that this heterozygosity is unlikely to explain the difference in basal antibody titers.

as explained above, after correction of the statistical analysis, IgA seems unchanged in both *Mb1Hells^{KO}* and *CD21Hells^{KO}*

- while IgG2c is significantly reduced only in *CD21Hells^{KO}*, we observe a similar trend in *Mb1Hells^{KO}*, with a high dispersion of the samples that may mask a true difference

- IgG3 is undoubtedly highly reduced in *Mb1Hells^{KO}* but preserved in *CD21Hells^{KO}*; nevertheless, as IgG3 is typically associated with T-independent responses, we think this difference may reflect differential Cre expression pattern that spares CD21^{low} B cells in CD21-Cre mice

The different impact of Hells loss-of-function on IgG3 basal titers between the two models is now underlined in the results part (lines 138-142).

We have also excluded the contribution of CD79 haploinsufficiency on other aspects of the phenotype such as pre-MBC enrichment and CD98 overexpression, by analyzing a third model, *Vav-iCre Hells^{KO}*, where the Cre recombinase is expressed throughout the hematopoietic compartment from the HSC stage (see figure below). **We did not intend to integrate these preliminary data in the present article, as we have not fully characterized this third model.**

***VaviHells^{KO}* recapitulate the pre-MBC and CD98 upregulation phenotype seen in *Mb1Hells^{KO}* GC B cells**

- (a) Representative FC plots showing CD95⁺GL-7⁺ and quantification of day 14 GC B cells in control (*n*=6) and *VaviHells^{KO}* (*n*=9) animals
- (b) Representative FC plots showing CCR6⁺CD38⁺ and quantification of pre-MBC population in control (*n*=6) and *VaviHells^{KO}* (*n*=9) animals
- (c) Representative CD98 staining of CD95⁺GL-7⁺CCR6^{neg}CD38^{neg} GC B cells and quantification of CD98 geometric MFI.

Bar charts and error bars represent the mean ± SD. Unpaired two-tailed *t*-test were performed for (a) to (c); ***: *p*≤0.001, ****: *p*≤0.0001.

2. The basis for the marked loss of GC B cells over time remains unclear given that there is no observed increase in apoptosis, and strongly reduced generation of mature memory B cells and plasma cells. The notable decrease in *FADS2* expression makes one wonder about increased ferroptosis. It would be helpful if a positive control for cells undergoing ferroptosis could be provided to ensure the assay is working.

A positive control had been performed for the ferroptosis experiment: we incubated splenocytes *in vitro* with various concentrations of the ferroptosis activator RSL3 and indeed observed a clear increase

in Bodipy-C11 fluorescence at 510 nm after treatment. **This control is now included in Supplementary Fig 6f, and described in the Methods section (lines 588-590).**

3. Another explanation for the reduction in GC B cells could be a reduced proliferation rate. Given the very rapid proliferation of GC B cells, small differences can lead to big effects. The data in Figure 4c show a trend toward more cKO cells in G1 and less in S. This analysis was done at day 10, when the effect on GC size seems variable. This analysis should be shown for a second, later time point.

We repeated the 2D cell cycle analysis on day 14 GC B cells and **present the new data in Fig. 4c and in the text (lines 190-193).** The former data on day 10 GC B have been moved to **Supplementary Fig. 4c.** We agree that these additional data strengthen the observation that HELLS loss-of-function does not affect cell cycle progression of GC B cells.

4. In figure 1 it is stated that Hells is overexpressed in GC B cells. This wording should be changed to indicate that Hells is more highly expressed in GC compared to naïve or follicular B cells. (Overexpression implies something non-physiological).

We changed the title of Fig. 1 and reformulated another sentence in the results part for the same reason (line 99).

5. Line 293, design should be designate.

We corrected this French false-friend (line 318).

REVIEWERS' COMMENTS

Reviewer #1 (Remarks to the Author):

The authors sincerely answered all of my questions. Now the manuscript is acceptable.

Reviewer #2 (Remarks to the Author):

I am satisfied that the revised manuscript and comments from the authors satisfy all of my concerns. Congratulations on a very exciting study and manuscript!

Reviewer #3 (Remarks to the Author):

The authors have adequately addressed my earlier concerns through their revisions.

Reviewer #1 (Remarks to the Author):

The authors sincerely answered all of my questions. Now the manuscript is acceptable.

Reviewer #2 (Remarks to the Author):

I am satisfied that the revised manuscript and comments from the authors satisfy all of my concerns. Congratulations on a very exciting study and manuscript!

Reviewer #3 (Remarks to the Author):

The authors have adequately addressed my earlier concerns through their revisions.

We are pleased to learn that the reviewers are satisfied with the changes introduced in the manuscript and with our previous responses. Once again, we would like to thank the reviewers for their appreciation of our work and for their comments which helped us improve the quality of the manuscript.

Yours sincerely

Sébastien Storck, Jean-Claude Weill and Claude-Agnès Reynaud